# A degron-mimicking molecular glue drives CRBN homo-dimerization and degradation

Gerasimos Langousis[1], Pablo Gainza [1], Moritz Hunkeler [2], Despoina Kapsitidou[1], Etienne J. Donckele[1], Stefano Annunziato[1], Lars Wiedmer[1], Katherine F. M. Jones[1], Bradley DeMarco[1], Chao Quan [1], Richard D. Bunker[1], Kevin J. Lumb [1], Bernhard Fasching[1], John C. Castle[1], Sharon A. Townson [1] & Débora Bonenfant[1] ✉

Cereblon (CRBN) is an E3 ubiquitin ligase widely harnessed for targeted protein degradation (TPD). We report the discovery of a molecular glue degrader (MGD), MRT-31619, that drives homo-dimerization of CRBN and promotes its fast, potent, and selective degradation by the ubiquitin proteasome system. Interestingly, the cryo-electron microscopy (cryo-EM) structure of the CRBN homodimer reveals a unique mechanism whereby two molecular glues assemble into a helix-like structure and drive ternary complex formation by mimicking a neosubstrate G-loop degron. This CRBN chemical knockout offers a valuable tool to elucidate the molecular mechanism of MGDs, to investigate its endogenous substrates and understand their physiological roles.

Cellular protein homeostasis relies heavily on the ubiquitin-proteasome system (UPS). The UPS cascade orchestrates ubiquitin chain formation on proteins destined for degradation by the proteasome[1]. Funneling to the UPS is often governed by substrate receptors that mediate ubiquitin transfer to cognate substrates at cullin-RING ligase (CRL) complexes. Cereblon (CRBN) is considered a prototypical substrate receptor that operates in the CRL4[CRBN] complex through an interaction with the adaptor DNA-damage binding protein 1 (DDB1)[2,3]. Initially associated with brain biology, CRBN research has uncovered its role in maintaining cell homeostasis[4] and has driven the advancement of targeted protein degradation (TPD) in clinical applications[5,6]. TPD mainly leverages compounds such as molecular glue degraders (MGDs) or proteolysis-targeting chimeras (PROTACs); many such compounds bind to the CRL4-CRBN ubiquitin ligase and promote recruitment of neosubstrates for ubiquitination and degradation[7]. CRBN PROTACs are heterobifunctional molecules composed of two covalently linked ligands, one for the neosubstrate and one for CRBN, coupled by a linker. CRBN MGDs are smaller molecules compared to PROTACS that bind directly to CRBN and modify the surface to enable recognition of a neosubstrate. The initially discovered CRBN-MGD complexes bind to G-loops, a structural motif on the neosubstrate's surface[8]. Such CRBN-MGD-neosubstrate ternary complexes are commonly stabilized by compound-protein and protein-protein interactions[9]. Recent studies have shown that MGD-bound CRBN can recognize other motifs beyond the G-loop, suggesting that the CRBN neosubstrate spectrum may be much larger than previously thought[10].

Here, we show the identification and characterization of MRT-31619, an MGD that drives CRBN degradation. Mechanistically, MRT-31619 mimics a G-loop degron and elicits CRBN-MRT-31619 dimerization with one CRBN molecule serving as a neosubstrate; this leads to CRBN ubiquitination and degradation by the UPS.

## Results

### MRT-31619 is a molecular glue degrader of CRBN

During proteomic profiling of our CRBN-focused compound library, we identified MRT-31619 (Fig. 1a) as a selective, fast, and potent inducer of CRBN degradation (Fig. 1b and Supplementary Fig. 1a). Previously, PROTACs have been employed to induce chemical knockdowns of E3 ligases, such as VHL homo-PROTACs[11,12], which induce VHL degradation as well as CRBN homo-PROTACs[13,14] and VHL-CRBN PROTACs[15,16], which both induce CRBN degradation. We compared MRT-31619 to the CRBN homo-PROTAC (OUN20985[13], Fig. 1a), noting that the compounds exhibit similar binary affinity to CRBN measured by biochemical TR-FRET thalidomide displacement assay (Supplementary Fig. 2). However, proteomics suggests that MRT-31619 outperforms

[1]Monte Rosa Therapeutics AG, Basel, Switzerland. [2]University of Basel, BioEM Lab, Basel, Switzerland. ✉e-mail: dbonenfant@monterosatx.com

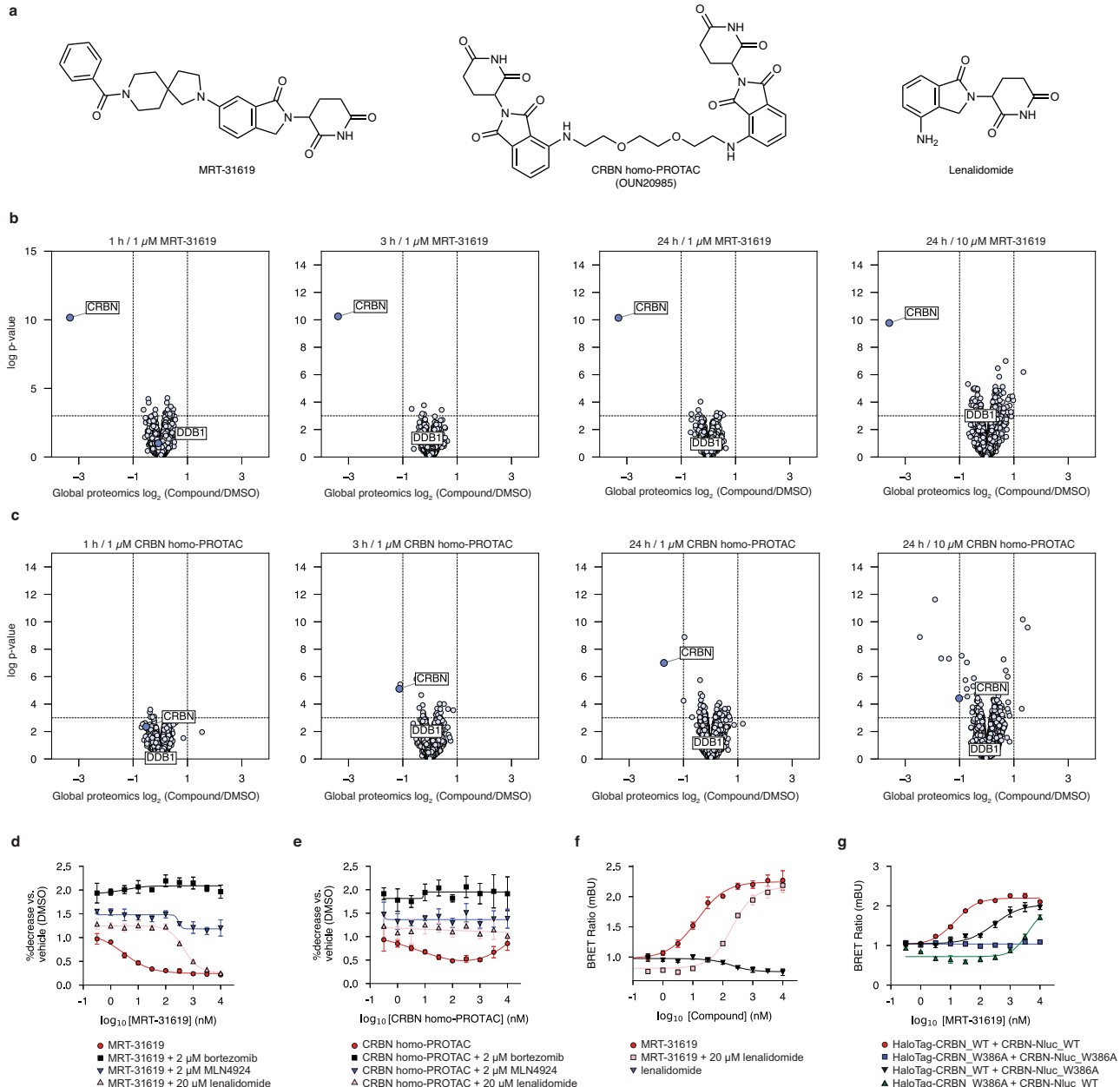

**Fig. 1 | MRT-31619 is a CRBN-dependent CRBN MGD. a** Chemical structure of MRT-31619, OUN20985 (CRBN homo-PROTAC) and Lenalidomide. **b, c** Global quantitative proteomics in Jurkat cells. Volcano plots show compound/DMSO protein abundance upon treatment with 1 μM of MRT-31619 or CRBN homo-PROTAC for 1 h, 3 h and 24 h, and with 10 μM of MRT-31619 or CRBN homo-PROTAC for 24 h. **d** HEK293T_HiBiT-CRBN cells were treated with MRT-31619 for 1 h. Bortezomib at 2 μM, MLN4924 at 2 μM, and lenalidomide at 20 μM were added 1 h prior to MRT-31619 treatment. Values were normalized to DMSO treatment. Four biological replicates were plotted as mean ± Standard Deviation (SD). **e** HEK293T_HiBiT-CRBN cells were treated with CRBN homo-PROTAC for 1 h. Bortezomib at 2 μM, MLN4924 at 2 μM, and lenalidomide at 20 μM were added 1 h

prior to CRBN homo-PROTAC treatment. Values were normalized to DMSO treatment. Four biological replicates were plotted as mean ± SD. **f** NanoBRET assay for HaloTag-CRBN and CRBN-NanoLuc constructs upon MRT-31619 or lenalidomide treatment. In the double treatment condition, lenalidomide was added at 20 μM, 1 h prior to MRT-31619 treatment. Values were normalized to DMSO treatment. Three biological replicates were plotted as mean ± Standard Error of Measurement (SEM). **g** NanoBRET assay for HaloTag-CRBN and CRBN-NanoLuc (CRBN-Nluc) wild-type (WT) and W386A constructs upon MRT-31619 treatment. Values were normalized to DMSO treatment. Three biological replicates were plotted as mean ± SEM. Source data are provided as a Source Data file.

the CRBN homo-PROTAC in terms of potency, selectivity and speed (Fig. 1b, c). Notably, after 1 h and up to 24 h at 1 μM, MRT-31619 remains highly effective in inducing degradation compared to the corresponding CRBN homo-PROTAC. Furthermore, MRT-31619 exhibits an excellent selectivity profile at concentrations of 1 μM and 10 μM for 24 h, whereas several known CRBN neosubstrates are degraded by the CRBN homo-PROTAC (ZFP91, ZNF692, ZNF653, RNF166, IKZF1, Supplementary Fig. 1b)[17–19]. Interestingly, MRT-31619 did not induce

degradation of DDB1 (Fig. 1b). We conclude that MRT-31619 is a potent and selective CRBN degrader.

Competition with the CRBN-dependent MGD lenalidomide (Fig. 1a) reduced MRT-31619-driven CRBN degradation (Fig. 1d), suggesting that the two compounds bind the same pocket. Additionally, we examined whether CRBN degradation occurs through the UPS. The degradation of CRBN induced by MRT-31619 was confirmed to be dependent on CRL and the proteasome. This was demonstrated by co-

treating cells with both MRT-31619 and the NEDD8 E1 inhibitor MLN4924 or bortezomib, respectively (Fig. 1d). These data indicate that MRT-31619 binds CRBN and promotes its degradation by the UPS.

Remarkably, MRT-31619 activity did not decrease at high compound concentrations (*hook effect*)[20]. The hook effect is a biochemical phenomenon observed at high PROTAC concentrations where the pockets on both sides of the interface become occupied by a ligand, preventing ternary complex formation. In contrast to MRT-31619, a hook effect is observed with the CRBN homo-PROTAC (Fig. 1e), while we confirmed the CRBN degradation rescue after co-treatment of CRBN homo-PROTAC with either lenalidomide, bortezomib or MLN4924 (Fig. 1e). Collectively, we show that MRT-31619 degrades CRBN via a mechanism distinct from a heterobifunctional molecule.

To address whether MRT-31619 promotes CRBN homo-dimerization, we set up a NanoBRET assay using HaloTag-CRBN and CRBN-NanoLuc constructs. We observed that MRT-31619 drives CRBN-CRBN interaction that is diminished upon lenalidomide competition (Fig. 1f). As a control, single lenalidomide treatment did not drive NanoBRET engagement, suggesting that occupying the tri-tryptophan (tri-Trp) pocket is not sufficient for CRBN-CRBN complex formation. Strikingly, MRT-31619 also induced an in vitro complex indicative of a CRBN-DDB1 dimer as shown by SEC-MALS analysis of purified recombinant proteins (Supplementary Fig. 3). Given that lenalidomide reduced MRT-31619 activity, we hypothesized that MRT-31619 binds to the tri-Trp pocket of the CRBN CULT domain formed by W380, W386 and W400. To test this, we used CRBN W386A[21] mutant constructs in our NanoBRET assay. In agreement with our hypothesis, CRBN-CRBN interaction was fully abrogated in the dual CRBN W386A condition (Fig. 1g). Collectively, these data indicate that MRT-31619 binds CRBN in the tri-Trp pocket and promotes CRBN homo-dimerization. We conclude that MRT-31619 is a bona fide MGD that makes CRBN a CRBN-dependent neosubstrate.

## CRBN dimerization is induced by two molecules of MRT-31619
We employed single-particle cryogenic electron microscopy (cryo-EM) to determine the structural basis of the homo-dimeric CRBN complex. Using purified recombinant CRBN-DDB1$^{\Delta BPB}$ (DDB1 lacking the beta-propeller domain beta) in the presence of MRT-31619, we obtained two maps at 2.9 Å and 3.1 Å resolution, respectively (Supplementary Fig. 4). The two reconstructions represent the major conformations resulting from the CRBN kink motion (Supplementary Fig. 5 and Supplementary Movie 1) and were both used for model building (Supplementary Table 1). No symmetry was imposed during image processing, as substantial conformational flexibility was identified in the complex (Supplementary movie 1). While both maps reveal a dimer of CRBN-DDB1$^{\Delta BPB}$ dimeric complexes, one DDB1$^{\Delta BPB}$ molecule is poorly resolved due to local flexibility and was omitted from the final models. Given its higher resolution (2.9 Å), we describe in detail conformation 1 (Fig. 2a).

Both protomers adopt the CRBN closed conformation[22] and interact as expected with DDB1$^{\Delta BPB}$. The sole interface between the interacting dimers is the CRBN neosurface comprising the MRT-31619-bound CULT domain and the N-terminal LON domain (Fig. 2b). MRT-31619 is bound in the tri-Trp pocket with the glutarimide moiety interacting with cis CRBN H378 and W380 through three canonical hydrogen bonds (Fig. 2c), consistent with binding interactions reported by other CRBN-binding ligands[23]. Moreover, the carbonyl group of the isoindolinone is at a distance for a weak hydrogen bond to the amine group of cis CRBN N351. Surprisingly, the two molecules of MRT-31619 adopt a complementary helix-like conformation that bridges the two CRBN tri-Trp pockets (Fig. 2b). The conformation of the two compound copies is virtually identical within an all-atom RMSD of 0.064 Å. The helix-like MRT-31619 dimer is stabilized by hydrophobic interactions between its spirocyclic linker regions. The rigid spirocyclic ring system forms a unique non-coplanar bicyclic architecture, in which a single carbon atom links pyrrolidine and

piperidine moieties. This spiro-conformation imposes a fixed dihedral angle of 81° between the two planes defined by the ring systems. The resulting conformational restriction leads to well-defined conformers in a helical arrangement, which also makes them shape-complementary to each other. The CRBN-bound compound conformations are further stabilized through multiple intermolecular interactions. Notably, hydrophobic contacts between the spirocyclic linkers contribute to dimer stabilization, and a putative non-classical C−H⋯O aromatic hydrogen bond is observed between the phenyl ring on one conformer and the carbonyl oxygen of the isoindolinone moiety of the other conformer. These cooperative interactions contribute to the observed structural integrity of the homo-dimeric complex, emphasizing the potential of spirocyclic frameworks to drive both conformational control and molecular recognition in ligand design.

The formation of the complex results in the burial of 1528 Å$^2$ of the solvent accessible surface area of both protomers (Fig. 2d), an area larger than most transient protein-protein interactions (800-1200 Å$^2$) and within the range for obligate homodimers[24]. We use the solvent-excluded surface (SES or molecular surface) to compute the footprint of the interface (see Methods). The footprint of protomer 1 shows a surface area of 455 Å$^2$, of which 109 Å$^2$ corresponds to the MGD (Fig. 2d). The footprint of these surfaces exceeds the typical size of transient protein interactions (typically 400 Å$^2$,[25]), and the footprint size of known CRBN neosubstrates (Supplementary Fig. 6)[9].

As the majority of the interface's footprint involves CRBN aminoacid residues, we investigated the contribution of the protein-protein contacts to the overall formation and stability of the complex. We focused specifically on residues not involved in the interaction with the MGD in either cis- or trans-, namely F150, F102, E377, R373, and V388 (Fig. 2d, g). Mouse CRBN differs from human CRBN in three of these residues, with the corresponding mutations F102S, V388I, and E377V. Treatment of mouse cells with either 1 μM or 10 μM of MRT-31619 did not show any reduction in CRBN levels (Supplementary Fig. 7). This was further confirmed in NanoBRET where there was no detectable interaction between mouse CRBN and human CRBN (Fig. 2e). However, mutating I391V and V380E (V388, E377 in human CRBN) or S105F (F102 in human) restores binding fully or partially, respectively, to levels similar to wild-type human CRBN (Fig. 2e). In contrast, mutating residues F150, R373 or E377 to alanine individually does not disrupt the CRBN-CRBN interaction in NanoBRET (Fig. 2f and Supplementary Fig. 8a−c). This suggests that ternary complex formation is primarily driven by compound-compound, compound-trans CRBN interactions, and secondarily by protein-protein interactions.

## MRT-31619 is a degron-mimicking CRBN molecular glue
In the cryo-EM structure, the MRT-31619 phenyl carbonyl moiety forms a hydrogen bond with the trans CRBN W400 side chain indole nitrogen (Figs. 2c and 3a). The CRBN-CK1α-lenalidomide structure (PDB: 5fqd[8]) shows the CK1α G-loop degron at positions N39 (side chain) and G40 (backbone) interacting with CRBN W400 side chain indole nitrogen (Fig. 3b). Superposition of the CRBN-CRBN structure with CRBN-CK1α-lenalidomide shows that MRT-31619 closely resembles the surface of the CK1α G-loop degron (Fig. 3c) and its essential hydrogen bond to W400. To elucidate the importance of compound-trans CRBN interactions, a series of rationally designed analogs were synthesized and evaluated (Fig. 3d−f).

MRT-31015 substitutes the terminal phenyl moiety of MRT-31619 with an acetyl group, thereby removing the aromatic group that mimics the surface of residue N39 in CK1α (Fig. 3e). In contrast, MRT-30568 retains the aromaticity through a benzyl group but lacks the carbonyl necessary to form the key hydrogen bond to CRBN W400 (Fig. 3f). Despite these structural alterations, MRT-31015 exhibits similar binary affinity to CRBN, as measured by a biochemical thalidomide CRBN displacement assay (Supplementary Fig. 2). In contrast,

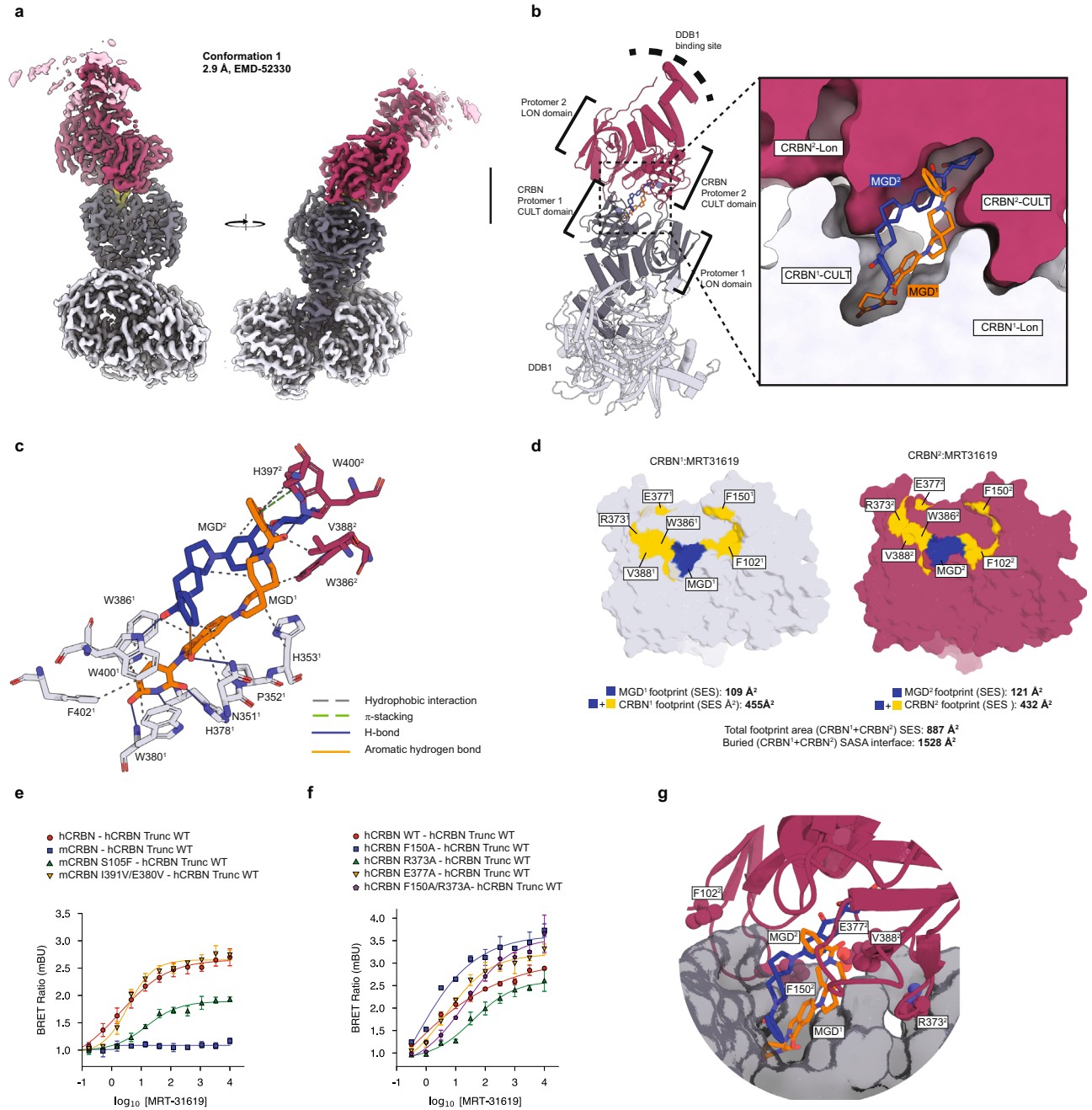

**Fig. 2 | CryoEM structure of the homo-dimeric CRBN complex. a** Cryo-EM density of conformation 1 at 2.9 Å, overall resolution (EMD-52330). **b** Model of DDB1 and the two CRBN protomers (CULT and LON domains) in cartoon representation. Overview and zoom-in at the tri-tryptophan CRBN pockets, illustrating how two molecules of MRT-31619 bridge the two CRBN subunits. **c** Zoom-in at the tri-tryptophan CRBN pockets, showing the MGD1–MGD2 interactions. **d** Footprints for CRBN-MRT-31619, protomer 1 and protomer 2: MRT-31619 footprints in blue and footprints of PPI in yellow. **e** NanoBRET assay for mouse CRBN mutants upon MRT-31619 treatment. Six biological replicates were plotted as mean ± SD. Human and mouse CRBN are indicated as hCRBN and mCRBN, respectively. Wild type (WT) and mutant HaloTag-CRBN constructs are indicated. For this experiment, hCRBN-NanoLuc constructs with a 191-248 aa deletion (Trunc) that do not bind DDB1 were used. **f** NanoBRET assay for hCRBN mutants upon MRT-31619 treatment. Three biological replicates were plotted as mean ± SEM. Wild type (WT) and mutant HaloTag-hCRBN and hCRBN-NanoLuc constructs are indicated. For this experiment, hCRBN-NanoLuc constructs with a 191-248 aa deletion (Trunc) that do not bind DDB1 were used. **g** Zoom-in at the tri-tryptophan CRBN pockets, showing key aminoacid interactions. Source data are provided as a Source Data file.

MRT-30568 shows slightly less affinity ($IC_{50} = 60$ nM) compared to MRT-31619 ($IC_{50} = 14$ nM).

Proteomics, NanoBRET, HiBiT and Western blot experiments were performed to compare the activity of MRT-31619 with analogs. MRT-31015 treatment led to weaker CRBN degradation compared to MRT-31619 (Fig. 3d, e, g and Supplementary Fig. 9), while MRT-30568 showed no degradation activity (Fig. 3d, f, g and Supplementary Fig. 9).

Additionally, MRT-31015 treatment resulted in reduced CRBN-CRBN NanoBRET interaction compared to MRT-31619, while MRT-30568 fully abrogated binding (Fig. 3h). We also conducted biophysical characterizations of MRT-31619 and its chemical analogs to evaluate the formation of the CRBN:MGD:CRBN ternary complex using CRBN-DDB1 purified recombinant proteins. Flow-induced Dispersion Analysis (FIDA) experiments confirmed that MRT-31015 treatment resulted in

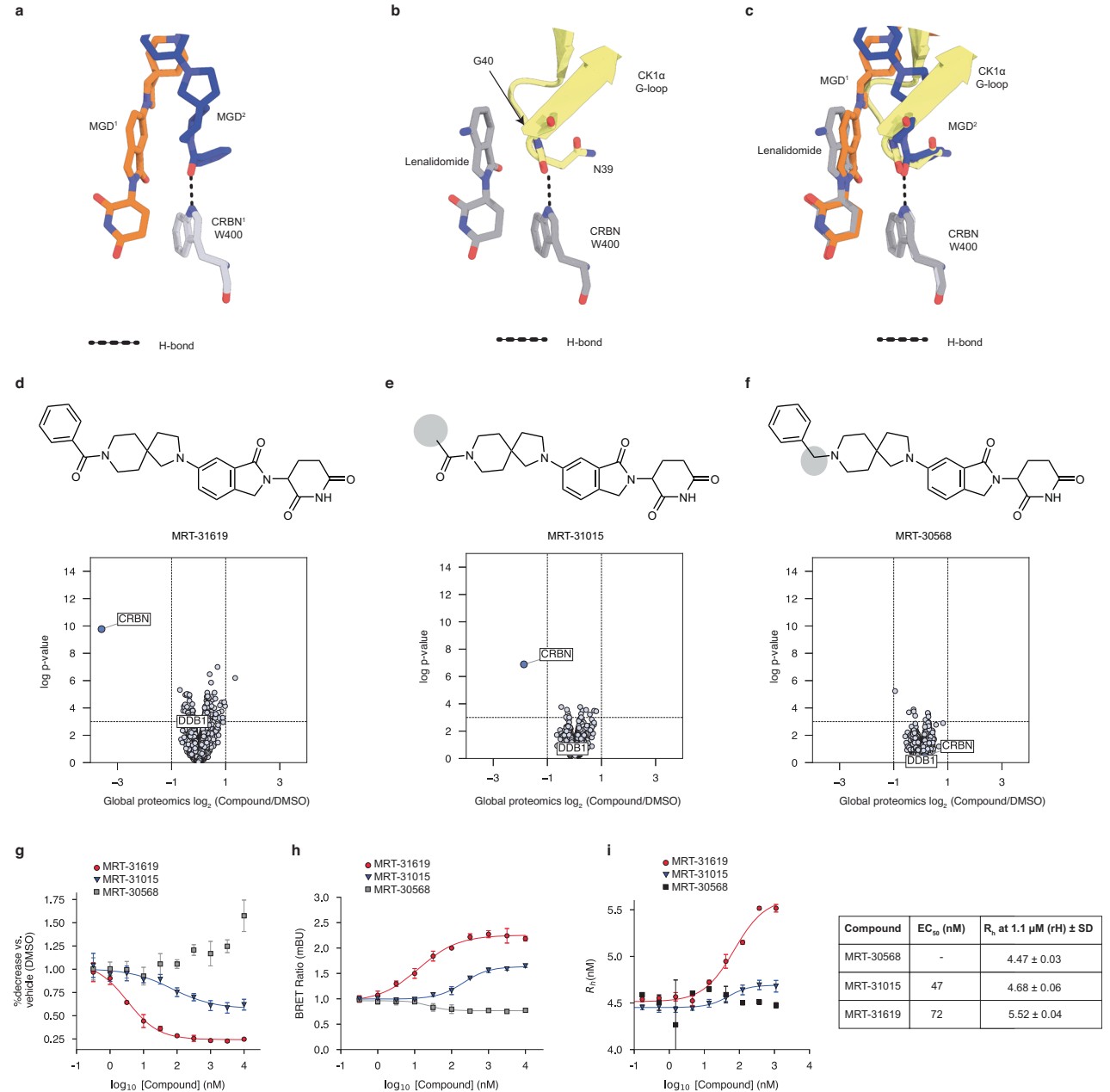

**Fig. 3 | MRT-31619 is a degron-mimicking CRBN molecular glue. a** The cis CRBN (CRBN[1]) binds one MRT-31619 molecule (MGD[1]) at the tri-tryptophan pocket and interacts with the trans MRT-31619 molecule (MGD[2]) via a hydrogen bond to W400. **b** CRBN W400 interacts with the CK1α G-loop via a hydrogen bond in the presence of lenalidomide. **c** Superposition of panels **a** and **b** indicates that MGD[2] is a structural mimic of the CK1α G-loop and employs the same hydrogen bond. **d–f** Chemical structures of MRT-31619, MRT-31015, or MRT-30568 and their corresponding global quantitative proteomics data in Jurkat cells. Volcano plots show compound/DMSO protein abundance upon treatment with 10 μM of MRT-31619,

MRT-31015, or MRT-30568 for 24 h. **g** HEK293T_HiBiT-CRBN degradation upon MRT-31619, MRT-31015, or MRT-30568 treatment for 1 h. Four biological replicates were plotted as mean ± SD. **h** NanoBRET assay for HaloTag-CRBN and CRBN-NanoLuc constructs upon MRT-31619, MRT-31015, or MRT-30568 treatment. Three biological replicates were plotted as mean ± SEM. **i** Flow-induced dispersion analysis (FIDA) for MRT-31619, MRT-31015, or MRT-30568 and the corresponding EC$_{50}$ and Max $R_h$ values. The predicted value of $R_h$ calculated from the cryo-EM structure of the ternary complex is 7.47 nm. Three biological replicates were plotted as mean ± SD. Source data are provided as a Source Data file.

reduced ternary complex formation compared to MRT-31619, whereas MRT-30568 did not show complex formation activity (Fig. 3i). Collectively, these data indicate that cellular recruitment affinity and the extent of biophysical ternary complex formation of MRT-31619 and its analogs correlate with the cellular potency of their respective degradation. These findings demonstrate that the key hydrogen bond between the carbonyl of MRT-31619 and trans CRBN W400 reflects a conserved G-loop degron interaction, and that compound potency is critically dependent on this structural mimicry.

## MRT-31619 ubiquitinates CRBN lysines near the DDB1 binding site

To further characterize the mode of action of MRT-31619, we conducted a global ubiquitinomics study to quantify intracellular lysine ubiquitination at endogenous levels in cells following MRT-31619 treatment. The mass spectrometry-based ubiquitinomics experiment identified twelve CRBN lysine ubiquitination sites that were strongly upregulated in MRT-31619-treated cells (Fig. 4a). CRBN has twenty-eight lysines in total, with all of them showing some surface

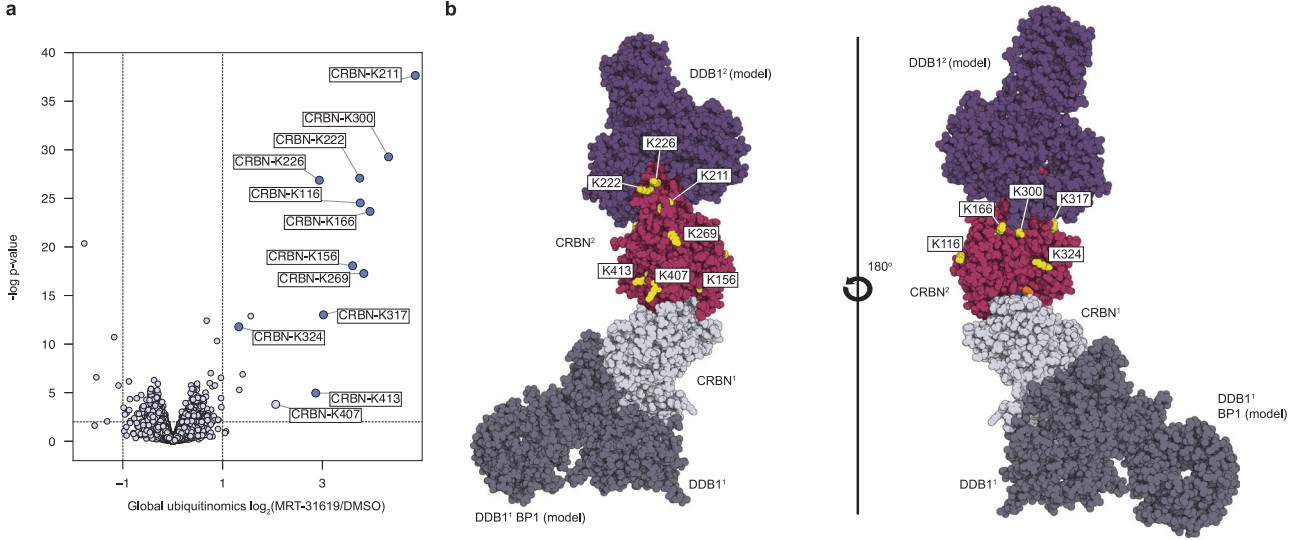

**Fig. 4 | Model and CRBN ubiquitinomics data. a** Global ubiquitinomics in Jurkat cells at 10 μM of MRT-31619 for 30 min. **b** Model for MRT-31619-driven CRBN ubiquitination at the CRL4CRBN complex.

accessibility in monomeric DDB1-free CRBN. We observe that nearly half of them are modified by ubiquitin after MRT-31619 treatment. The most affected lysine sites on CRBN are K211, K300, K222, and K166, showing up 30 times more enrichment compared to untreated cells (Fig. 4a). Interestingly, these lysines are located in an area adjacent to and partially occluded by DDB1 (Fig. 4b), and K300 is displayed on the opposite face of CRBN. Although several lysine residues on DDB1 are in close proximity to CRBN, only K823 exhibited a significant increase in ubiquitination following MRT-31619 treatment (Supplementary Fig. 10). However, this modification at K823 was insufficient to induce DDB1 degradation, as confirmed by global proteomics analysis (Fig. 1b).

Given the proximity of the most ubiquitinated lysines of CRBN to the DDB1 binding site (Supplementary Fig. 11), the lack of significant DDB1 degradation and poor ubiquitination remains an open question. One possibility is that the position of DDB1 is less optimal for ubiquitin transfer than CRBN. A second possibility is that DDB1-free CRBN is the most prevalent ubiquitinated form, possibly after ubiquitin is first added to CRBN, hindering DDB1 binding, which would explain why lysines that are adjacent to the DDB1 binding site are the most ubiquitinated. In addition to these two hypotheses, several other factors, such as E3/E2 dynamics and protein stoichiometry, could help explain the lack of DDB1 degradation. The precise mechanism by which DDB1 is spared while CRBN is potently degraded remains enigmatic, yet our structural and ubiquitination data provide a valuable model system for future studies on the dynamics and behavior of the UPS.

## Discussion

The increasing sensitivity and high-throughput capacity of mass spectrometry-based proteomics[26–28], combined with expanding libraries of structurally diverse CRBN-based molecular glues[9], is enabling the discovery of unconventional degron recognition mechanisms, without prior assumptions about binding modes. In this study, an unbiased proteomic screen revealed MRT-31619 as a fast, potent, and selective CRBN degrader. Since protein downregulation can result from various indirect effects, we corroborated our findings through a combination of biophysical, biochemical, cellular, and structural orthogonal methods. These analyses demonstrated that, unlike CRBN molecular glues that facilitate CRBN-substrate interactions, MRT-31619 induces CRBN homo-dimerization through a composite interface driven by compound-compound, protein-protein, and compound-trans-CRBN interactions. The latter mimics a G-loop-like

degron in trans, ultimately triggering CRBN's own degradation by the ubiquitin-proteasome system. This self-targeting mechanism redefines CRBN as both an E3 ligase and a neosubstrate, revealing a degron-mimicking mechanism and expanding the conceptual scope of molecular glue recognition.

MRT-31619 can serve as a chemical knock-out tool to deplete CRBN in cells, for instance, in competition assays to assess the CRBN dependency of other molecular glues. In contrast to classical genetic approaches, the small-molecule MRT-31619 offers practical advantages such as facile application to diverse cell and tissue types, speed, and reversibility. Moreover, MRT-31619 should be superior to current CRBN-targeting PROTACs[13–16] since it enables rapid, potent and selective CRBN degradation and does not rely on the activity of other E3 ligases.

Given its qualities, MRT-31619 represents a valuable chemical probe for interrogating CRBN biology. CRBN is expressed across various tissues and has been linked to multiple cellular processes and diseases by targeting proteins for degradation through the ubiquitin-proteasome system[29]. Multiple endogenous CRBN substrates have been described in the literature[23,30–32]. In addition, an emerging body of work has demonstrated that CRBN is involved in maintaining cellular homeostasis through the recognition of a C-terminal cyclic imide, which may occur spontaneously[4] or catalyzed by other proteins[33]. Interestingly, recent studies[34–36] showed a potential link between CRBN and neurodegeneration, suggesting that CRBN could be a potential therapeutic target for preventing neurological disorders. The selective and acute knockout of CRBN by MRT-31619 offers a valuable tool to investigate CRBN's endogenous substrates and define its functions in human neurodegenerative models.

A distinct finding in this study is the discovery of a small-molecule-induced CRBN homodimer stabilized by two molecules of MRT-31619. This finding echoes earlier work in the protein-protein interaction field that showed that as few as one point mutation on protein surfaces can induce homo-oligomerization[37]. Small molecules that induce protein-protein homo-dimerization have been previously reported, for example, in the context of KRAS[38], resulting in distinct biological effects compared to non-dimerizing inhibitors. Our findings provide a mechanistic basis that may inform the rational design of homo-dimerizers: small molecules that bind near native interfaces could be engineered through solvent-exposed groups to promote helix-like packing and mimic protein interactions. This approach offers a potential strategy for modulating biological pathways via enforced homo-oligomer formation.

Overall, MRT-31619 is a versatile tool for inducing CRBN degradation, investigating its endogenous substrates and physiological roles, elucidating the molecular mechanisms of MGDs, and potentially guiding the rational design of small molecules that induce protein homo-oligomerization.

## Methods

### Mass spectrometry-based methods

Jurkat WT cells were cultured in RPMI-1640 medium (Gibco Cat#A10491-01) plus 10% FBS (PAN Biotech Cat#P30-109) at 37 °C with 5% $CO_2$. Cells were seeded (3-4 million cells) in 6-well plates and treated for either 1 h, 3 h or 24 h at 1 µM or 10 µM with MRT-31619 (see Supplementary Methods) or CRBN Homo-PROTAC (Selleckchem Cat#S2881) or DMSO (Sigma Cat# 41639) and for 24 h at 10 µM with MRT-31015 (see Supplementary Methods) or MRT-30568 (see Supplementary Methods) or DMSO. J774A.1 Mouse cells were cultured in DMEM medium (Gibco Cat#11965-092) plus 10% FBS (PAN Biotech Cat#P30-109) at 37 °C with 5% CO2. Cells were seeded (2 million cells) in 6-well plates and treated for either 1 h or 24 h at 1 µM or 10 µM with MRT-31619, CRBN Homo-PROTAC, MRT-31015, MRT-30568 or DMSO. MRT-31619, MRT-31015, and MRT-30568 were part of a CRBN-focused compound library comprising approximately 50,000 molecules (see Supplementary Methods).

Compound and DMSO-treated conditions were prepared in biological duplicates. Cells were collected in 2 mL Eppendorf Lobind tubes and were pelleted by centrifugation at $250 \times g$ for 5 min at 4 °C. Supernatants were removed, cells were transferred to 1.5 mL Eppendorf Lobind tubes and were washed with 1 mL of PBS. After centrifugation, cell pellets were flash-frozen on dry ice and stored at −80 °C. Cell pellets were lysed using 80 µL PreOmics iST-NHS (Cat#P.O.00030) lysis buffer. A tryptophan assay was used to determine lysate concentration. Plates were read with a Tecan Infinite 200 PRO, and lysates were then normalized to 90 µg in 50 µL. The normalized lysates were then processed using the recommended PreOmics sample preparation protocol with minor modifications. In brief, proteins were reduced, alkylated, and digested for 3 h at 37 °C. The peptides were then labeled with tandem mass tag (TMT) reagent (1:4; peptide:TMT label, Cat#A44520, A52046) for 1 h at 22 °C. Labeled peptides were then quenched with 5% hydroxylamine solution diluted in LC/MS-grade water. The peptides from the 16 and 18 conditions were then combined in a 1:1 ratio and purified with PreOmics desalting cartridges. Mixed and labeled peptides were subjected to high-pH reversed-phase HPLC fractionation on an Agilent X-bridge C18 column (3.5 µm particles, 2.1 mm I.D, and 15 cm in length, Cat#186003023). Using an Agilent 1200 LC system, a 60 min linear gradient from 0% to 40% acetonitrile in 10 mM ammonium formate, pH 10, separated the peptide mixture into a total of 60 fractions, which were then consolidated into 24 fractions. Peptide concentrations were estimated by UV, and after drying in a SpeedVac integrated vacuum concentrator, all 24 fractions were resuspended to 0.2 µg/µL with 0.1% formic acid based on the peptide UV chromatogram. Each of the fractions was loaded onto a 25 cm Aurora Ion Optics column (75 µm I.D., 1.6 µm particles, Cat#AUR3-25075C18) using a Vanquish Neo HPLC system (Thermo Fisher Scientific). The peptides were separated using a 168 min gradient from 4% to 30% buffer B (80% acetonitrile in 0.1% formic acid) equilibrated with buffer A (0.1% formic acid) at a flow rate of 400 nL/min. Eluted TMT peptides were analyzed using a Thermo Fisher Scientific Orbitrap Eclipse mass spectrometer.

MS1 scans were acquired at a resolution of 120,000 with 400–1400 mass over charge (m/z) scan range, Automatic Gain Control (AGC) target $4 \times 10^5$, maximum injection time 50 ms. Then, MS2 precursors were isolated using the quadrupole (0.7 m/z window) with AGC $1 \times 10^4$ and maximum injection time 50 ms. Precursors were fragmented by collision-induced dissociation (CID) at a normalized collision energy (NCE) of 35% and analyzed in the ion trap. Following MS2, synchronous precursor selection (SPS) MS3 scans were collected by using high-energy collision-induced dissociation (HCD), and fragments were analyzed using the Orbitrap (NCE 55%, AGC target $1 \times 10^5$, maximum injection time 120 ms, resolution 60,000). Protein identification and quantification were performed using Proteome Discoverer 2.5 with the SEQUEST algorithm and UniProt human database (SwissProt, release 2019, UP000005640_9606, 20602 protein sequences) or the UniProt mouse database (UP000000589_10090, 55,387 protein sequences). Mass tolerance was set at 10 ppm for precursors and at 0.6 Da for fragments. A maximum of 2 missed cleavages was allowed. Methionine oxidation was set as a dynamic modification, while TMT tags on peptide N termini/lysine residues and cysteine alkylation (+113.084) were set as static modifications. Adjustment of reporter ion intensities for isotopic impurities according to the manufacturer's instructions was performed in Proteome Discoverer. Subsequently, protein identifications were inferred from unique peptides, i.e., peptides matching multiple protein entries were excluded. Protein relative quantification was performed using an in-house developed software in Python and R. This analysis included multiple steps; de-duplication of PSMs to peptide charge modification (PCM) level using the average, the sum and the maximum abundance of TMT channels to sort and pick the best PCM, addition of 1 to abundances of PCMs (for all channels) if any of the abundance values was below 0 (for subsequent log2 transformation). MSstatsTMT[39,40] was used for global normalization of TMT channels (equalize medians), to normalize PCMs and summarize protein abundances using Tukey's Median Polish. Missing values were imputed using maxQuantileforCensored=NULL. As an intermediate step, before protein summarization, a custom algorithm was applied to find outliers of imputed values (cases where the model fails), and all PCMs for the given protein were reset to being missing. Proteins with at least 2 peptide features are displayed on the volcano plots. An arbitrary L2FC cut-off of −1 and +1 was applied to the dataset corresponding to a minimal −50% depletion (x0.5) or +100% enrichment (x2), respectively. The p-value cut-off for statistical significance was set at the standard $10^{-2}$ value.

### Ubiquitinomics analysis

Jurkat cells (10 million cells/sample) were treated with the indicated compounds for 30 min, followed by washing with PBS (on ice). Compound and DMSO-treated conditions were prepared in biological quadruplicate. The cell pellets were resuspended in sodium deoxycholate (SDC) buffer containing 0.5% SDC, 10 mM TCEP, 40 mM CAA, 75 mM Tris-HCl at pH = 8.5[41]. The lysates were heated to 90 °C for 10 min while shaking at 1000 rpm (rotations per minute). Protein concentrations were determined using the BCA assay (Merck-Millipore, Cat#71285) and the proteins were digested overnight at 37 °C using 100:1 protein:trypsin ratio (Promega, Cat#V5071). After digestion, immunoprecipitation (IP) buffer (50 mM MOPS pH 7.2, 10 mM $Na_2HPO_4$, 50 mM NaCl) was added to the samples together with K-GG antibody-bead conjugate (CST, Cat#5562), followed by a 2 h incubation on a rotor wheel. Beads washing and peptide elution were performed according to the manufacturer's instructions. The peptide eluate was desalted using in-house prepared, 200 µL two plug C18 StageTips (3 M EMPORE™).

Peptides were loaded on 35 cm reversed-phase columns (75 µm inner diameter, packed in-house with C18-AQ 1.9 µm resin [ReproSil-Pur®, Dr. Maisch GmbH]). The column temperature was maintained at 55 °C using a column oven. A Vanquish Neo system (Thermo Fisher Scientific) was directly coupled online with the mass spectrometer (timsTOF Ultra, Bruker) via a nano-electrospray source. The LC flow rate was 300 nL/min and the complete gradient 45 min (ubiquitinomics). Data acquisition was done using slicePASEF mode[42].

MS raw files were analyzed using DIA-NN (1.9.1)[43]. Reviewed UniProt entries (human, SwissProt 10-2022 [9606]) were used as the protein sequence database for DIA-NN searches. One missed cleavage

and a maximum of two variable modifications (N-terminal acetylation and K-GG (UniMod: 121)) were allowed. N-terminal excision of methionine was enabled. Carbamidomethylation of cysteines was set as a fixed modification. The data processing was carried out using library-free analysis mode in DIA-NN. --tims-scan was added as an additional command.

DIA-NN outputs were further processed with R. Peptide precursor quantifications with missing values in more than 50% of samples, or <25% of compound-treated samples, were discarded. K-GG peptide abundances were calculated based on precursor ion intensity levels using the MaxLFQ algorithm[44], as implemented in the DIA-NN R package (https://github.com/vdemichev/diann-rpackage/). K-GG peptide to site mapping was done using reviewed entries of the human UniProt database (SwissProt, release 10-2022). The peptide intensities were normalized by median scaling and corrected for variance drift over time (if present) using the principal components (derived from principal component analysis (PCA)) belonging to DMSO samples. Subsequently, protein (or peptide) intensities were subjected to statistical testing with variance and log fold-change moderation using LIMMA (PMID: 25605792). $p$-values corrected for multiple testing were used to assess significance for ($q$-value < 0.05).

### HiBiT degradation assay

HEK293T (obtained from ATCC) CRBN-knockout expressing HiBiT[45]-CRBN (driven by the human phosphoglycerate kinase 1 promoter) cells were cultured in DMEM medium (Gibco Cat#10569-010) plus 10% FBS (PAN Biotech, Cat#P30-109) at 37 °C, 5% $CO_2$. Cells were collected and resuspended in DMEM medium lacking phenol red (Gibco Cat#21063-029) plus 10% FBS, 1 mM sodium pyruvate (Gibco #11360-070), 1× penicillin-streptomycin (Gibco Cat#15140122) at $3 \times 10^5$ cells/mL. Cells were then seeded (25 μL, 7500 cells per well) in a 384-well low flange white flat bottom plate (Corning Cat#3570) and incubated overnight at 37 °C, 5% $CO_2$. Cells were treated for 1 h with DMSO (Sigma Cat#41639) at 0.1% final concentration and MRT compounds or homo-PROTAC CRBN degrader (Selleckchem Cat#S2881) at serial dilution using a Tecan D300e Digital Dispenser. When applicable, cells were pre-treated for 1 h with bortezomib (Selleckchem Cat#S1013), MLN4924 (Selleckchem Cat#S7109) or lenalidomide (Selleckchem Cat#S1029). Nano-Glo® HiBiT Lytic Reagent was prepared according to the manufacturer's instructions (Promega Cat# N3030), and 25 μL were dispensed to each well. The Plate was incubated for 10 min at room temperature, and luminescence was detected with the PHERAstar FSX plate reader (BMG LABTECH). Signal was normalized to DMSO treatment.

### Computation of PPI footprints and buried areas

To compute PPI footprints and their areas, we used the solvent-excluded surface area, also known as the molecular surface[46]. Specifically, we used the definition of buried surfaces (i.e., the interface of a PPI) identical to that presented in ref. 9. The buried surface of a subunit (e.g., one copy of CRBN-DDB1 with one copy of MRT-31619) in a complex structure (i.e., the cryoEM structure) was calculated by comparing surface vertices of the surface mesh of the subunit and the surface mesh of the complex. Vertices in the subunit's mesh were labeled as 'interface' meshes if more than 2.0 Å away from any complex surface vertex.

### NanoBRET assay

HEK293T WT cells were cultured in DMEM medium (Gibco Cat#10569-010) plus 10% FBS (PAN Biotech, P30-109) at 37 °C, 5% $CO_2$. Cells were collected and resuspended in DMEM medium lacking phenol red (Gibco Cat#21063-029) plus 10% FBS, 1 mM sodium pyruvate (Gibco Cat#11360-070), 1× penicillin-streptomycin (Gibco Cat#15140122) at $4 \times 10^5$ cells/mL. Cells were seeded (2 mL, $8 \times 10^5$ cells per well) in a 6-well plate (Corning Cat#CLS3516) and incubated overnight at 37 °C,

5% $CO_2$. Cells were transfected with HaloTag-CRBN and CRBN-NanoLuc plasmids at 10:1 ratio (2 μg:200 ng) using FuGENE® HD Transfection Reagent (Promega Cat#E2311) and incubated overnight at 37 °C, 5% $CO_2$. For certain experiments, a CRBN deletion 191-248 aa NanoLuc construct that lacks the DDB1-interacting helices[47] was used. Cells were collected, resuspended in DMEM medium lacking phenol red (Gibco Cat#21063-029) plus 10% FBS, 1 mM sodium pyruvate (Gibco Cat#11360-070), 1× penicillin-streptomycin (Gibco Cat# 15140122) at $1 \times 10^5$ cells/mL and seeded (25 μL, 2500 cells per well) in a 384-well low flange white flat bottom plate (Corning Cat#3570), treated (1:1000 dilution) with HaloTag® NanoBRET® 618 Ligand and incubated overnight at 37 °C, 5% $CO_2$. Cells were treated for 3 h with DMSO (Sigma Cat#41639) at 0.1% final concentration, MRT compounds or lenalidomide (Selleckchem Cat#S1029) at serial dilution using a Tecan D300e Digital Dispenser. When applicable, cells were pre-treated for 1 h with lenalidomide (Selleckchem Cat#S1029). NanoBRET® Nano-Glo® substrate was added, and donor/acceptor signals were measured according to manufacturer instructions (Promega Cat#N1661) at the PHERAstar FSX plate reader (BMG LABTECH). The NanoBRET corrected ratio, including background subtraction and DMSO normalization, was calculated.

### Biochemical thalidomide displacement assay

Compound activity for displacing thalidomide from CRBN was monitored in a TR-FRET assay using a Cy5-functionalized thalidomide tracer as a fluorescent probe. Assays were conducted in Greiner white 384-well HiBase plates (Cat#784075-25) in 10 μL total volume. A one-pot detection solution of 6xHis-CRBN-DDB1 complex (2.5 nM), anti-His Terbium Cryptate Gold (1X, PerkinElmer Cat# 61HI2TLB), and Cy5-Thalidomide (100 nM, Tenova Cat#T52461) was prepared in 20 mM HEPES, 20 mM NaCl, 0.2 mM TCEP, 0.2 mM EDTA, and 0.005% Tween20. Compounds were stored in dry, ambient temperatures at 10 mM. An 11-point, 1:3 dilution series was prepared from 1 mM stock concentrations in Echo-compatible LDV plates using a Bravo liquid handler (Agilent, USA). 200 nL of each compound dilution series was dispensed into assay wells using an Echo 650 (Labcyte Inc., USA). 200 nL of 2 mM CC-885 (Medchem Express Cat# HY-101488) was transferred into the active-control wells for the assay, and 200 nL of DMSO was transferred into the neutral-control wells. The assay was then allowed to incubate for 45 min at ambient temperature after transferring the compound. Plate measurements were taken on an Envision (Revvity, USA) using the red filter (Ex. 337 nm, em1: 615 nm, em2: 665 nm) (Flashes: 50, Integration time: 60–400 μs, Z-height: 10 mm, Ratio-multiplier: 1). The TR-FRET signal was then subsequently normalized to the neutral and active controls. Analysis and $IC_{50}$ values were derived using KNIME analytics (KNIME AG, Switzerland) transformation and fitting within Prism 10.4.1 (GraphPad Software, USA) using a four-parameter logistic fitting model. $K_i$ was derived from the geometric mean of the $IC_{50}$ values using the Cheng-Prusoff transformation.

### Western blot protocol for CRBN degradation

Jurkat WT cells obtained from ATCC were cultured in RPMI-1640 medium (Gibco Cat#A10491-01) plus 10% FBS (PAN Biotech Cat#P30-109) at 37 °C with 5% $CO_2$. Cells were seeded (2 million cells/well) in 12-well plates and treated for 1 h, 3 h or 24 h with 1 μM or 10 μM compound or DMSO (Sigma Cat#41639). Cells were collected in 2 mL Eppendorf Lobind tubes and were pelleted by centrifugation at 300 rcf for 5 min at 4 °C. Supernatants were removed, and cells were washed with 1 mL of PBS. After centrifugation, cell pellets were flash-frozen on dry ice and stored at −80 °C. Cell pellets were lysed for 30 min with 45 μL RIPA lysis buffer (Thermo Fisher Cat#89900) supplemented with protease inhibitor cocktail (Sigma Cat#P8340), Phosphatase Inhibitor solution II (Sigma Cat#P5726), Phosphatase Inhibitor solution III (Sigma Cat#P0044) and Pierce Universal Nuclease for Cell Lysis

(Thermo Cat# 88701). A BCA assay (Pierce™ Rapid Gold BCA Protein Assay Kit, Thermo Cat#A55861) was used to determine lysate concentration. The plate was read with a PHERAstar FSX, and lysate concentrations were then normalized. Samples were analyzed by western blot using the following primary antibody, incubated overnight at 4 °C, diluted 1/1000: CRBN, Novus Biologicals, NBP1-91810. Membranes were incubated for 1 h at room temperature with a secondary antibody, anti-rabbit HRP antibody diluted 1/2500 (Invitrogen, Cat. No. A16110), and GAPDH, typically used as a loading control (Invitrogen, Cat#MA5-15738-A647). Membranes were developed on a ChemiDoc imaging system (Biorad) using ECL (enhanced chemiluminescence) substrates (Biorad).

## SEC-MALS analysis
Purified recombinant CRBN-DDB1 complex was diluted with 10 mM HEPES pH 7.1, 100 mM NaCl, 0.5 mM TCEP to 9 µM final concentration. MRT-31619 (or equivalent amount of DMSO) was added to 10 µM final concentration, and samples were incubated for 60 min at 4 °C. Samples were centrifuged at 12500 rcf, 3 min, 4 °C before injection into Superdex Increase 200 10/300 (0.5 mL/min flow rate) coupled to Agilent Infinity 1260 HPLC with three-angle MALS detectors and RI detector (Wyatt Technology). Chromatography data analysis and determination of the molecular mass were performed with ASTRA 8.1.2.1 software (Wyatt Technology).

## Flow-induced dispersion analysis
Fida 1 instrument with 480 nm LED fluorescence detection was used for binding experiments (Fida Biosystems ApS) with a FIDA standard capillary (i.d.: 75 µm, LT: 100 cm, leff: 84 cm). An analyte solution of 250 nM 6xHis-CRBN(E40-442L)/DDB1(M1-1140H) complex was prepared in 20 mM HEPES pH 8.0, 150 mM NaCl, 0.05% Pluronic acid F127, and 0.5 mM TCEP. An indicator solution of 50 nM Tris-NTA OG488 (ATT Bioquest, Cat#12615) and 250 nM 6xHis-CRBN(E40-442L)/DDB1(M1-1140H) complex was prepared in 20 mM HEPES pH 8.0, 150 mM NaCl, 0.05% pluronic acid F127, and 0.5 mM TCEP. An 11-point, 1:3 dilution series of the compound was prepared from 10 mM stock in a 384-well Echo LDV plate in 100% DMSO. Compound (100 nL) was transferred to a dry 96-well PP V-bottom plate in duplicate using an Echo 665 (Beckman). 100 µL of analyte solution and 100 µL of indicator solution were added to each replicate, respectively. The FIDA system sample tray was held at 25 °C until the injection. A standard FIDA capillary was rinsed with 1 M sodium hydroxide at 3500 mBar for 45 s, then equilibrated in assay buffer at 3500 mBar for 75 s. A 20 s injection at 3500 mBar of analyte solution was performed to fill the capillary, indicator solution was then injected for 10 s at 50 mBar. The indicator solution was then mobilized with the analyte solution and measured with a 180 s injection at 400 mBar at 25 °C. This injection scheme was repeated in technical triplicate for each compound treatment condition. Data were analyzed in the Fida software (3.1) and plotted in GraphPad Prism (10.4.1). $EC_{50}$ values were determined using the [Inhibitor] versus response – variable slope (four parameters) nonlinear fitting model (GraphPad Software, USA).

## Recombinant protein expression and purification
Strep-TEV tagged CRBN[41-442] and untagged full-length DDB1 used for SEC-MALS, or Strep-TEV tagged CRBN[41-442] and untagged DDB1[ΔBPB] (amino acids 1-395-GNGNSG-706-1140) used for cryo-EM grids were cloned into pFastBac1 vectors. Recombinant baculovirus stocks were prepared using Bac-to-Bac Baculovirus Expression System (Thermo Fisher Scientific) and were used to infect Sf9 insect cells for co-expression of CRBN/DDB1 or CRBN/DDB1[ΔBPB]. Harvested insect cells were resuspended in lysis buffer containing 100 mM Tris-HCl, pH 8.0, 150 mM NaCl, 1 mM TCEP, 5% glycerol, 1 mM PMSF, 1 tablet cOmplete EDTA-free protein inhibitor (Roche) and 250 units/L Benzonase and homogenized. The cell lysate was cleared by centrifuging

at 23,000 rcf at 4 °C for 90 min. The clarified supernatant was filtered through a 0.22 µm membrane filter and loaded onto a Strep-Tactin column (IBA Lifesciences) pre-equilibrated with the binding buffer containing 50 mM Tris-HCl, pH 8.0, 150 mM NaCl and 1 mM EDTA. The column was washed, and the proteins were eluted with a buffer containing 50 mM Tris-HCl, pH 8.0, 150 mM NaCl, 1 mM EDTA and 50 mM D-Biotin. The eluates were desalted into a buffer supplemented with 10% v/v glycerol and cleaved with TEV protease at 4 °C overnight. The TEV reaction was loaded onto a Strep-Tactin column to remove the strep-tag and the uncleaved fraction, followed by dilution into 5 times the volume of buffer containing 50 mM MES pH 6.5, 5% glycerol and 1 mM TCEP. The diluted protein was loaded onto a Q-HP column and eluted using a gradient of NaCl up to 1 M. The eluate was loaded onto a Superdex 200 16/600 pg column equilibrated with 10 mM HEPES, pH 7.4, 150 mM NaCl and 1 mM TCEP. Fractions corresponding to soluble CRBN/DDB1 or CRBN/DDB1[ΔBPB] were concentrated to 30-40 mg/mL, aliquoted, flash-frozen, and stored at −80 °C.

## Cryo-EM sample preparation and data acquisition
Purified recombinant CRBN[41-442]-DDB1[ΔBPB] were diluted to 10 mg/mL in a buffer containing 20 mM HEPES pH 7.3, 100 mM NaCl, 0.5 mM TCEP and 100 µM of MRT-31619. The sample was incubated on ice for 30 min and was then clarified by centrifugation at 13,000 rcf for 5 min at 4 °C. The sample was kept on ice and n-dodecyl-β-D-maltoside (DDM, Sigma) was added, resulting in final concentrations of 0.001% DDM, 9 mg/mL CRBN[41-442]-DDB1[ΔBPB] and 90 µM MRT-31619, and incubated for 1 min. 4 µL were applied to a glow-discharged (20 mA, 60 sec, Quorum GloQube, UK) holey carbon-coated grid (Quantifoil R2/1, 300 mesh (Au)). The grids were blotted for 3 s with Whatman 1 filter paper and vitrified in liquid ethane at −180 °C using a Leica EM GP2 plunger (Leica Microsystems, Austria) operated at 90% humidity and 10 °C. Vitrified grids were clipped and transferred into a Titan Krios G4 electron microscope (Thermo Fisher Scientific, USA) operating at 300 kV and equipped with a Falcon4i direct electron detector and a Selectris X energy filter (10 eV slit width). 20,582 movies (52 frames each) were collected using the aberration-free image shift scheme implemented in EPU (3.8.1) at a magnification of 165,000x (resulting in 0.73 Å/pix on the specimen level), with a total electron exposure of 40 e-/Å². The defocus was varied between −0.8 µm and −2 µm throughout the data collection.

## Data processing and model building
All data processing was performed in Relion (5.0)[48], unless otherwise mentioned. All resolutions are given based on the FSC = 0.143 threshold criterion[49,50]. The dataset was split into four batches to speed up calculations. The batches were corrected for stage drift and beam-induced motion, and the image defocus values were estimated using CtfFind (4.1.10)[51] in 4 × 4 patches. Only movies with a CTF resolution of better than 5 Å (20,275 movies) were kept for subsequent analysis. TOPAZ[52] (0.2.5) was used with the general model to pick 4,802,227 particles, which were extracted at a box size of 352 × 352 pixels (257 × 257 Å), and Fourier-cropped to a box of 128 × 128 pixel size (resulting in 2.0 Å/pix). The particles were cleaned through a total of 4 rounds of 2D classification, starting with 4 batches and gradually combining classes that displayed secondary structure features. 768,180 particles (representing the heterodimer of heterodimers) were sent into 3D classification (4 classes, $\tau = 4$) with alignment. One class was selected (442,229 particles) and the particles were used for DynaMight[53] flexibility analysis (Supplementary Movie 1). The supplementary movie represents a circular journey through the latent space, following the white-to-red gradient indicated in Supplementary Fig. 4. The particles were re-extracted in a box of 384 × 384 pixels (280 × 280 Å) and Fourier-cropped to 1.1 Å/pix (256 × 256 pixels box size). The signal-to-noise ratio of the particles was increased through

one round of Bayesian polishing[54], followed by CTF refinement[55] correcting for higher-order aberrations and anisotropic magnification. Per-particle defocus values were determined, and the resulting particles yielded a consensus reconstruction at 2.9 Å resolution. Masked classifications with a mask around two CRBN subunits (without alignment, $\tau = 100$) led to the identification of two major conformations at 2.9 Å and 3.1 Å (Supplementary Fig. 2), representing the extremes of the kink motion of CRBN. A third class (resembling conformation 1) was identified at this step. Particles contained in this class led to a worse overall map with no additional insights, and it was thus not followed up in detail. All maps were post-processed in Relion and with deepEMhancer[56] (0.16), and all maps were used for model building. Blush regularization was used throughout all 3D classifications and refinements. Local resolution ranges are given based on the 0–70% in local resolution histograms.

ChimeraX (1.8)[57] was used to initially place two molecules each of DDB1 and CRBN (taken from 8TNQ[58]) into the density, but the DDB1 subunit with the worst density was subsequently removed, leaving one DDB1 and two CRBN molecules in the structure. This initial model was relaxed into the density using ISOLDE (1.7)[59] prior to manual inspection and model building in COOT (0.9)[60]. The ligand was placed, and the model was prepared for refinement with phenix.ready_set (1.21_5207)[61] and then sent through phenix.real_space_refine (1.21_5207)[62], first against the unsharpened map (refining ADPs) and then against the sharpened map from Relion (without ADP refinement). Final model validation was performed against the sharpened map. All maps and models were deposited in the EMDB and PDB under accession codes EMDB-52330/PDB-9HPI (conformation 1) and EMDB-52331/PDB-9HPJ (conformation 2). Data collection parameters and refinement statistics can be found in Supplementary Table 1.

### Reporting summary

Further information on research design is available in the Nature Portfolio Reporting Summary linked to this article.

## Data availability

Proteomics data were deposited to the ProteomeXchange (http://www.proteomexchange.org/) with identifiers PXD060261, PXD060544 and PXD065335 (https://www.ebi.ac.uk/pride/archive/projects/PXD060261, https://www.ebi.ac.uk/pride/archive/projects/PXD060544, https://www.ebi.ac.uk/pride/archive/projects/PXD065335). The cryo-EM maps were deposited in the Electron Microscopy Data Bank (EMDB) database under accession numbers (conformation 1) EMD-52330 and (conformation 2) EMD-52331. The atomic models were deposited in the Protein Data Bank (PDB) database under the accession codes (conformation 1) PDB 9HPI and conformation 2 PDB 9HPJ. Source data for data plots in the figures are provided at: https://doi.org/10.6084/m9.figshare.29424770 and with this paper as a Source Data file. Source data are provided with this paper.

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

## Acknowledgements

We are grateful to the MRT Proteomics team. We thank the MRT Discovery team and Rémi Fertin for the HitBiT-CRBN cell line and the MRT structural biology team for valuable discussions. We thank Gian Marco de Donatis for assistance in setting up NanoBRET/HiBiT assays. We thank Viva Biotech for protein purification. We thank the BioEM lab of the Biozentrum at the University of Basel for support with sample screening and data collection. We also thank sciCORE (University of Basel) for assistance with high-performance computing. We thank NEOsphere Biotechnologies for providing ubiquitinomics data. We thank 2bind GmbH for the SEC-MALS analysis.

## Author contributions

G.L., P.G., and D.B. conceived the project. G.L. designed all experiments, collected proteomics samples and performed NanoBRET and HiBiT experiments. S.A. performed the mouse NanoBRET experiments. D.K. performed proteomics experiments. K.F.M.J. and B.D. performed the FIDA experiments. M.H. performed the cryo-EM analysis and built atomic models. C.Q., R.D.B., L.W. and P.G. analyzed structures. E.J.D. and B.F. designed chemical compounds. G.L., P.G., and M.H. prepared figures. G.L. and P.G. prepared the initial manuscript with input from all authors. P.G. and D.B. revised and expanded the manuscript. K.J.L., S.A.T., J.C.C. and D.B. supervised research.

## Competing interests

P.G., D.K., E.J.D., S.A., L.W., K.F.M.J., B.D., C.Q., K.J.L., B.F., J.C.C., S.A.T., and D.B. are current employees and shareholders of Monte Rosa Therapeutics. G.L. and R.D.B. are former employees of Monte Rosa Therapeutics. The remaining author declares no competing interests.
