## [Transparent Peer Review file · Nature Communications]

A degron-mimicking molecular glue drives CRBN homo-dimerization and degradation

Corresponding Author: Dr Débora Bonenfant

Version 0:

Reviewer comments:

Reviewer #1

(Remarks to the Author)

The authors identified MRT-31619 as a CRBN degrader, a curious surprising finding since the compound was originally designed as a conventional molecular glue to recruit and degrade other neo-substrates to CRBN (not CRBN to itself!). It is often curiosity and surprising unexpected findings that drives innovation and discovery, and the authors are commended here for pursuing this finding despite clearly taking them off-track on their company based drug discovery goals.

The authors use a proteomics approach to compare MRT-31619 with the CRBN homoPROTAC 15a (here called Homo-PROTAC degrader 1), previously published by Steinebach et al. in 2018. The authors use a proteomics approach in Jurkat cells and show that MRT-31619 mediates selective degradation of CRBN at 1 μ M after 1 hour, 3 hours and 24 hours. In this experiment they also show that neither MRT-31619 nor the CRBN homoPROTAC 15a degrade DDB1.

To investigate whether degradation of CRBN with MRT-31619 is ubiquitin-proteasome system dependent, they demonstrate that CRBN is more ubiquitinated in the presence of MRT-31619 compared to a DMSO control using a global ubiquitinomics method in Jurkat cells. Using a HEK293 cell line expressing HiBit-tagged CRBN, they show that CRBN degradation mediated by MRT-31619 is proteasome- and cullin RING-dependent. They also highlight that CRBN can be rescued by lenalidomide at lower MRT-31619 concentrations. The authors qualify MRT-31619 as a molecular glue degrader, rather than a heterobifunctional degrader, supported by evidence that MRT-31619 does not appear to have a hook effect.

the real star of the show here is the cryo-EM structure, with the other results supporting this as a main findings. Despite conformational heterogeneity, the authors refine two consensus volumes representing two distinct conformations, which are overall of good-quality. Given the flexibility, one DDB1 molecule is poorly resolved, and the authors wisely omit this from final models. Remarkably, from cryo-EM it appears that not one, but two molecules of MRT-31619 mediate the CRBN dimerisation, with the two molecules wrapping around each other in a helical-like arrangement. Interestingly, the dimerisation of CRBN appears to be driven by interactions between MRT-31619 and CRBN, and also interactions between the two molecules of MRT-31619 via the spiro-cyclic linkers, and only minor protein-protein interactions between the two CRBN moieties are apparent.

The authors design and produce two further analogues of MRT-31619. MRT-31015 lacks the phenyl moiety which was shown by cryo-EM to pi-stack with CRBN H397, and this compound showed reduced CRBN degradation by proteomics at 10 μ M after 24 hours treatment and reduced CRBN-dimerisation by NanoBRET. MRT-30568 retains the phenyl moiety, but lacks the carbonyl group of MRT-31015 which the authors showed by cryo-EM could hydrogen-bond with CRBN W400 nitrogen side-chain, mimicking the Gly-loop interaction. The authors show by proteomics that the MRT-30568 analogue does not degrade CRBN at 10 μ M after 24 hours treatment and does not form a CRBN dimer by NanoBRET.

Together, the study builds a picture of how a small molecule degrader designed as conventional molecular glue recruits and dimerises two molecules of CRBN. The results shown are compelling, particularly with the cryo-EM structures revealing the binding mode and further mechanistic studies validating the mode of action. However, several comparisons on compound ternary complex stability and degrader potency are made, based on single assays performed at single timepoints or single concentrations. To this end, We would encourage the authors to consider performing additional orthogonal assays to verify their current data and unambiguously demonstrate the stability and potency of their degrader molecule relative to their control compounds and the previously published CRBN homoPROTAC 15a. For examples, studies to measure ternary

complex cooperativity and stability (e.g. via direct binding assays using ITC and/or SPR, or displacement binding assays vs a fluorescent CRBN-binding fluorescent probe would seem ideal).

Overall, the paper is significant and reports an exciting unexpected and wholly relevant finding, as it reveals an unprecedented mechanism by which a molecular degrader non-covalently auto-dimerises, to, in turn, dimerise CRBN. It also reveals how the benzaldehyde-group of one molecule of MRT-31619 can mimick CK1a G40 loop hydrogen-bonding the CRBN W400 in the presence of lenalidomide. This is an interesting and impactful study which, in our opinions, should eventually be published subject to some revisions outlined below.

Comments:

1) Page 1 lines 60-62. MRT-31619 is compared with CRBN PROTAC in terms of selectivity, potency, and rate of degradation.

The authors state "Proteomic comparison to a CRBN homo-PROTAC, hereafter referred to as CRBN PROTAC (Extended data Fig. 1), revealed that MRT-31619 is superior in terms of degradation, speed and potency (Fig. 1a)." it is difficult to draw strong comparisons on potency based on a single concentration. The Methods state that 1 μ M and 10 μ M concentrations were used, however the manuscript only shows data for 1 μ M. It would be worthwhile showing the 10 μ M treatment in the extended data.

Similarly, if any other timepoints or compound concentration treatments were performed for the analogues MRT-31015 and MRT-30568, these should be included in the supplementary information to rule out any possibility of hook effect or CRBN recovery coming into play at this higher concentrations and longer timepoint.

2) The authors should describe how they produced the sample for cryo-EM grid application. If the protein sample was taken directly from the SEC-MALS experiment described in the Methods, this should be specified.

3) Could the authors comment on any interactions between the two molecules of MRT-31619 bound in the complex??

4) In line 82, the authors describe the in vitro complex between MRT-31619 and CRBN-DDB1 as stable, however the stable character of this complex is not fully described by this single measurement at a single concentration. At 9 μ M CRBN-DDB1 and 10 μ M MRT-31619, only partial dimerisation of CRBN-DDB1 is observed. The authors could consider refraining from making claims on stability in this case.

5) To further characterise the ability of MRT-31619 to mediate dimerisation of CRBN, compared to the CRBN homoPROTAC 15a previously published by Steinebach et al. in 2018, and compared to the chemical analogues MRT-31015 and MRT-30568 published in this study, the authors could consider performing further biochemical or biophysical characterisation. Several examples of assays are described in Maniaci et al. Nat. Commun. 2017 (DOI: 10.1038/s41467-017-00954-1), Diehl et al. Angew. Chem., 2024 (DOI: 10.1002/anie.202319456) which should be cited. Given that more stable ternary complexes with degrader molecules have been shown to drive degradation efficiency and potency, it would be interesting to understand here if CRBN dimerisation correlates with degradation efficacy.

6) It is unclear whether the maps used for validation are deepEMhancer or RELION post-processed maps. It is a widely accepted opinion that the modified densities from density-modifying AI model programs should be primarily treated as modelling aids. The reviewers encourage the authors to perform any final atomic refinements and validation processes using volumes which have not been modified with AI-based procedures such as deepEMhancer.

7) Could the authors specify which atomic models were used to generate the alignment for ubiquitination modelling in Extended Data Fig 10? From our own modelling using cryo-EM structures of Cul4A-DDB1 in the presence/absence of an E2 (PDB codes: 8B3G and 7OKQ, aligning DDB1 with the cryo-EM structures in this study), several lysine residues from the ubiquitinomics results shown in Fig. 1(b) on the trans CRBN protomer appear to be well-poised for ubiquitination, even in the presence of another molecule of DDB1. No lysines on DDB1 were shown to be ubiquitinated in this system, this is consistent with our modelling which shows that DDB1 is outside of the reach of E2-Ub. Therefore, we disagree with the statement in line 129 that the second CRBN protomer is devoid of DDB1. We would also like to highlight that in Extended Data Fig 10, the binding site denoted for the E2 is likely to be incorrect, as shown by several published X-ray crystallography and cryo-EM structures of cullin RING and other RING E3 ligases engaging ubiquitin-loaded E2s.

8) This seems a very short article, with unusually only two figures. It feels a little sparse and the narrative could expand. Also many panels from the SI could be moved to main text. the paper seems to have been written in a rush, with narrative jumping up and down, e.g. ubiquitinomics data presented early, then referred to it later on when introducing the model? It is suggested that many figure panels and data to be moved to the main text, for example: Compounds chemical structure of key compound(s) should be included in main text figure Comparison with CRBN Homo-PROTAC degrader 1 (also called compound 15a in the original publication) should feature more prominently upfront in the main text

9) use of scholar language is encouraged throughout - e.g. use of word "catapulted" seems excessive.

10) "...there are minimal protein-protein interactions between the two CRBN protomers" - could quantify this? for example by calculating buried SASA within the ternary complex (see Cowan & Ciulli Ann. Rev. Biochem 2021 (PMID: 35320687, currently cited as ref. 3); and PMID: 33930325). on this point - the NanoBRET data with PPI mutations is important and should be quantified.

11) "...our data suggest a model where one CRBN protomer is part of a fully assembled CRL4 complex whereas the second protomer (the neosubstrate) is devoid of DDB1." this seems inconsistent with the SEC-MALS data which is consistent with a homo-dimeric complex of 2x CRBN-DDB1. It seems hard to imagine CRBN protein being stable alone and without being bound to DDB1.. what does a model of 2x full-length CRL4-CRBN look like?

12) reference list if OK, though could be expanded. For example, the authors might want to acknowledge and cite the original first homoPROTAC paper which was on VHL homo-dimerizers (PMID: 29018234)

13) EDF10: why not mapping all the K site found ubiquitinated in the MS spec ubiquitinomics experiment?

Minor points:

8) Page 1 line 49. We suggest re-wording "...these compounds bind to the CRBN CULT domain", as this makes it seem as though all MGDs and PROTACs recruit and bind the CRBN CULT domain.

9) Differences in nomenclature 'HiBit-CRBN' and 'CRBN-HiBit' make it unclear whether the HiBit is on the N- or C-terminus of CRBN.

10) In Extended Data Fig. 4, the identity of 'Cpd-1' in the figure legend should be indicated as MRT-31619. This should also be reflected in the Methods (line 429).

11) Page 2 line 82. "Dimeric CRBN-DDB1 dimer" should be rephrased.

12) In Extended Data Fig 5, could the authors briefly comment on what the composition of the grey-coloured volume was which constituted 25% of the 3D classification results?

13) The FSC curved in Extended Data Fig 5 (b) and (c) are difficult to read.

14) From the viewing distribution, it appears that the two structures solved have some preferential orientation, and thus the global and local resolutions by FSC could be inflated. The directional FSCs, as well as the sphericity value, should be included for both structures and the authors should consider using the 3D FSC server for FSC plots: <https://3dfsc.salk.edu>

15) For cryo-EM data validation, the authors should consider including all defocus values used, adding the 3DFSC sphericity value and EMRinger score.

16) Could the authors highlight the presence or absence of any symmetry in the MRT-31619 dimer assembly?

We agree to waive our anonymity in a spirit to enhance the transparency of the peer review process, Alessio Ciulli & Charlotte Crowe

Reviewer #2

(Remarks to the Author)

Reviewer #3

(Remarks to the Author)

This manuscript reports the discovery of a small molecule that induces dimerization and in turn degradation of cereblon by acting as a homodimer-inducing molecular glue. The authors discover the compound as selectively targeting cereblon by using proteomic profiling and present data in Jurkat cells showing high selectivity. The mechanism of interaction is resolved by using cryoelectron microscopy to solve the structure of cereblon with the small molecule (MRT-31619) and DDB1. The conclusions are supported by the data and manuscript well-written. Only two small suggestions are made below.

A figure should be included that shows the interactions between MRT-31619 and the tri-Trp pocket, as described at line 102-104.

It's not clear what the authors mean at line 135; are they proposing 'analogous pathway' to MRT-31619 induced dimerization and if so, how would that be induced endogenously?

Reviewer #4

(Remarks to the Author)

In this manuscript Langousis and coworkers reported a molecular glue MRT-31619 that mediates the dimerization and degradation of CRBN. Intriguingly, two MGD molecules assemble into a helix-like structure and connect two copies of CRBN together. This is an interesting and unique finding in MGD design. This study included detailed biochemical, structural and cellular experiments to demonstrate the binding mode of MRT-31619 to CRBN. The results and conclusion are convincing. Given the important role of CRBN in targeted protein degradation, the report of MRT-31619 as a new CRBN

molecular glue is well justified. Overall, this manuscript is well-written and concise, and should be of interest to a broad range of readers of the journal.

What is the binding affinity of MRT-31619 to CRBN? This information may be obtained using SPR or BLI binding assays using the small molecule and recombinant proteins.

Figure 2 should be improved to show more details of the MRT-31619 binding to CRBN by including more residues interacting with the small molecule.

Given that the two CRBN CULT domains demonstrated minimum direct interactions when being brought together by two MRT-31619 molecules, how rigid is the ternary complex of two CULT domains linked by the small molecules? Do the cryoEM images show particles with different conformations?

In the Extended Data Figure 9b, treatment with MRT-31619 led to degradation of CRBN. However, MRT-30568 treatment led to increase in CRBN level. Why?

In the model of CRL4CRBN complex bound with MRT-31619 as shown in the Extended Data Figure 10, the ubiquitination sites on the neosubstrate CRBN2 are facing away from the E2 binding site on CRL4. How would the charged E2 transfer the Ub to the lysine sites on CRBN?

What is the application of the MRT-31619 in targeted protein degradation if it removes CRBN from the treated cells. This should be discussed in the manuscript.

Version 1:

Reviewer comments:

Reviewer #1

(Remarks to the Author)

The authors have done an excellent good job at revising the manuscript. The degradation data has been strengthened by performing additional western blot experiments, and more robust comparison on degradation efficacy and mode of action in comparison to the previously published CRBN homoPROTAC has been brought up-front in the manuscript. The new compound has also been compared with the previously published CRBN homoPROTAC in terms of in-solution binding by TR-FRET, monitoring a fluorescently-labelled thalidomide probe displacement. We previously highlighted that information seemed unclear in terms of cryo-EM sample preparation, and the information has now been added to the manuscript. No changes have been made to the cryo-EM data, but the results have been further dissected by highlighting key interactions, interfaces and buried surfaces. Data on how the well-known human to mouse species residues variations impact on TC formation and compound-induced CRBN degradation has been added. Some further discussion has been added for the ubiquitinomics passage albeit the conclusions from this section remain somewhat limited. However we view this as acceptable to be included as is, with the due caveats and limitations.

Overall, the paper is significant and reports an exciting unexpected and relevant finding. We previously made several suggestions, and we are satisfied that these have been addressed. In the process of strengthening the manuscript, several sections have been altered, and additional experiments have been performed, meriting another complete read-through of the latest manuscript version. We believe this version is much improved and we recommend publication in the journal, subject to some additional and final revisions, as outlined below:

- 1) Line 75: and Supplementary Figure 2: for the 'biochemical thalidomide displacement assay' please specify in the main manuscript and in the figure legend that this is a TR-FRET assay between CRBN and thalidomide.
- 2) Supplementary Figure 2: does Rh refer to hydrodynamic radius? If so, is this a usual readout for a TR-FRET assay?
- 3) Line 108: based on the data shown in Fig. 1g, it does not appear that the CRBN-CRBN interaction is fully abrogated in the dual CRBN W386A condition (green curve). In contrast the HaloTag-CRBN(W386A) with CRBN(WT)-NanoLuc does not yield any increase in BRET ration upon compound treatment. Have these been mislabelled in the plot?
- 4) The authors state in Supplementary figure 10 that DDB1 is not ubiquitinated in the presence of MRT-31619, however in the main manuscript they state that DDB1 K823 showed significant ubiquitination. Given the stability of the CRBN-DDB1 complex, it seems unlikely that CRBN would exist without DDB1. Several other factors could come into play, such as CRBN conformational change in the presence of MRT-31619 and another CRBN molecule versus its apo form in the presence of DMSO, and also flexibility of CUL4 in the presence of two units of CUL4A-E2-Ub which would create steric hindrance and thus adapt the 'light face' of accessible lysines to the E2 active site cysteine.
- 5) In Supplementary figure 11. For clarity, the authors could consider colouring the lysine residues which showed evidence of ubiquitination in a different colour. Can any clusters of ubiquitinated lysines on CRBN be highlighted, as has been shown for example in Fig 2 (B) DOI: 10.1126/sciadv.ado6492?

6) When discussing the qualification and potential utility of their compound as a CRBN degrader, the authors should duly acknowledge that MRT-31619 is not the first CRBN-degrader that has been developed in the field, and thus duly reference the previous studies, not only the CRBN homo-PROTAC study (that is already cited in the manuscript as reference 13), and the RSC Med Chem 2021 (also cited as ref. 14), but also the papers that reported the VHL-CRBN ligase vs ligase PROTACs, which were all found to induce degradation of CRBN and not VHL – specifically, Girardini et al. Bioorg Med Chem, 2019 (PMID: 30826187), and Steinebach et al. Chem Commun 2019 (PMID: 30672516).

Minor points:

7) Line 70: 'During proteomic profiling of our compound CRBN library' should be rephrased, for example 'CRBN-focused compound library' or 'CRBN-recruiting compound library'

8) In Figure 1, the authors adequately explain in the figure legend what each abbreviation means (e.g. btzb, len, MLN, HT, etc.). However, there seems to be ample space to write out most of the full term below the figures themselves. This would be easier to understand, particularly for those outside of or new to the TPD field.

9) Some inconsistencies when referring to the CRBN homoPROTAC 15a: CRBN homo-PROTAC, CRBN-homoPROTAC, CRBN homoPROTAC and CRBN PROTAC are present in the main manuscript, figures and SI.

10) Figure 2e/f and Supplementary figure 8: the plots are difficult to read with all curves except the wild-type being in grey. Could the authors clarify in the figure legend the difference between hCRBN and mCRBN?

11) Supplementary figure 9 and lines 184-187: could the N = 2 replicate degradation western blots be quantified relative to GAPDH for clarity to the reader? For example, in the final lane (MRT-30568 10 uM 24 hours) the CRBN band appears to decrease in intensity, however GAPDH is also at lower intensity.

12) Could the authors clarify or rephrase the following sentence, lines 210-213: Despite the proximity of most ubiquitinated lysine, only one DDB1 ubiquitinated lysine at K823 showed significant upregulation after MRT-31919 (Supplementary Fig.10), which was insufficient to drive DDB1 degradation, as shown by proteomics (Fig.1b)

13) In the new Figure 4 (b), on the right-hand-side panel DDB1 has been written as "DDB".

We agree to waive our anonymity in a spirit to enhance the transparency of the peer review process, Alessio Ciulli & Charlotte Crowe.

Reviewer #2

(Remarks to the Author)

Reviewer #3

(Remarks to the Author)

This manuscript has been improved and my concerns have been addressed. One remaining concern is that there does not appear to be a PDB validation report for the deposited structures. It's possible it was present in the first submission and missing in the revision or that it's present and just eluding me.

Reviewer #4

(Remarks to the Author)

The authors have addressed my points satisfactorily. The manuscript has been substantially improved and is now ready for publication.

Response to reviewers for “A degron-mimicking molecular glue drives CRBN homo-dimerization and degradation” (ID#/NCOMMS-25-15248-T)

We thank the four reviewers for their thorough review of this manuscript, their enthusiasm for the work, and their encouraging comments. We also thank them for their constructive feedback, that has improved the clarity and quality of this manuscript. Please find our responses to the individual points below.

Reviewer #1 (Remarks to the Author):

The authors identified MRT-31619 as a CRBN degrader, a curious surprising finding since the compound was originally designed as a conventional molecular glue to recruit and degrade other neo-substrates to CRBN (not CRBN to itself!). It is often curiosity and surprising unexpected findings that drives innovation and discovery, and the authors are commended here for pursuing this finding despite clearly taking them off-track on their company based drug discovery goals.

The authors use a proteomics approach to compare MRT-31619 with the CRBN homoPROTAC 15a (here called Homo-PROTAC degrader 1), previously published by Steinebach et al. in 2018. The authors use a proteomics approach in Jurkat cells and show that MRT-31619 mediates selective degradation of CRBN at 1 uM after 1 hour, 3 hours and 24 hours. In this experiment they also show that neither MRT-31619 nor the CRBN homoPROTAC 15a degrade DDB1.

To investigate whether degradation of CRBN with MRT-31619 is ubiquitin-proteasome system dependent, they demonstrate that CRBN is more ubiquitinated in the presence of MRT-31619 compared to a DMSO control using a global ubiquitinomics method in Jurkat cells. Using a HEK293 cell line expressing HiBit-tagged CRBN, they show that CRBN degradation mediated by MRT-31619 is proteasome- and cullin RING-dependent. They also highlight that CRBN can be rescued by lenalidomide at lower MRT-31619 concentrations. The authors qualify MRT-31619 as a molecular glue degrader, rather than a heterobifunctional degrader, supported by evidence that MRT-31619 does not appear to have a hook effect.

the real star of the show here is the cryo-EM structure, with the other results supporting this as a main findings. Despite conformational heterogeneity, the

authors refine two consensus volumes representing two distinct conformations, which are overall of good-quality. Given the flexibility, one DDB1 molecule is poorly resolved, and the authors wisely omit this from final models. Remarkably, from cryo-EM it appears that not one, but two molecules of MRT-31619 mediate the CRBN dimerisation, with the two molecules wrapping around each other in a helical-like arrangement. Interestingly, the dimerisation of CRBN appears to be driven by interactions between MRT-31619 and CRBN, and also interactions between the two molecules of MRT-31619 via the spiro-cyclic linkers, and only minor protein-protein interactions between the two CRBN moieties are apparent.

The authors design and produce two further analogues of MRT-31619. MRT-31015 lacks the phenyl moiety which was shown by cryo-EM to pi-stack with CRBN H397, and this compound showed reduced CRBN degradation by proteomics at 10 μ M after 24 hours treatment and reduced CRBN-dimerisation by NanoBRET. MRT-30568 retains the phenyl moiety, but lacks the carbonyl group of MRT-31015 which the authors showed by cryo-EM could hydrogen-bond with CRBN W400 nitrogen side-chain, mimicking the Gly-loop interaction. The authors show by proteomics that the MRT-30568 analogue does not degrade CRBN at 10 μ M after 24 hours treatment and does not form a CRBN dimer by NanoBRET.

Together, the study builds a picture of how a small molecule degrader designed as conventional molecular glue recruits and dimerises two molecules of CRBN. The results shown are compelling, particularly with the cryo-EM structures revealing the binding mode and further mechanistic studies validating the mode of action. However, several comparisons on compound ternary complex stability and degrader potency are made, based on single assays performed at single timepoints or single concentrations. To this end, We would encourage the authors to consider performing additional orthogonal assays to verify their current data and unambiguously demonstrate the stability and potency of their degrader molecule relative to their control compounds and the previously published CRBN homoPROTAC 15a. For examples, studies to measure ternary complex cooperativity and stability (e.g. via direct binding assays using ITC and/or SPR, or displacement binding assays vs a fluorescent CRBN-binding fluorescent probe would seem ideal).

Overall, the paper is significant and reports an exciting unexpected and wholly relevant finding, as it reveals an unprecedented mechanism by which a molecular degrader non-covalently auto-dimerises, to, in turn, dimerise CRBN. It also reveals how the benzaldehyde-group of one molecule of MRT-31619 can mimic CK1a G40

loop hydrogen-bonding the CRBN W400 in the presence of lenalidomide. This is an interesting and impactful study which, in our opinions, should eventually be published subject to some revisions outlined below.

Response: We thank the Reviewer for this positive and accurate summary of our paper. We really appreciate the feedback.

Comments:

1) Page 1 lines 60-62. MRT-31619 is compared with CRBN PROTAC in terms of selectivity, potency, and rate of degradation.

The authors state “Proteomic comparison to a CRBN homo-PROTAC, hereafter referred to as CRBN PROTAC (Extended data Fig. 1), revealed that MRT-31619 is superior in terms of degradation, speed and potency (Fig. 1a).” it is difficult to draw strong comparisons on potency based on a single concentration. The Methods state that 1 μ M and 10 μ M concentrations were used, however the manuscript only shows data for 1 μ M. It would be worthwhile showing the 10 μ M treatment in the extended data.

Similarly, if any other timepoints or compound concentration treatments were performed for the analogues MRT-31015 and MRT-30568, these should be included in the supplementary information to rule out any possibility of hook effect or CRBN recovery coming into play at this higher concentrations and longer timepoint.

Response: We thank the reviewer for identifying these points. We have included in new Figure 1, two additional volcano plots: 10 μ M treatment at 24 h for both MRT-31619 (**Fig. 1b**) and the CRBN Homo-PROTAC (**Fig. 1c**). These additional data confirmed the potency in degradation of MRT-31619 compared to CRBN Homo-PROTAC. We have also modified the “**Mass spectrometry-based methods**” for additional clarification (line 272-276).

For MRT-31015 and MRT-30568, only a single proteomics experiment was conducted at 24 h with a concentration of 10 μ M (Fig. 3e and 3f). Additionally, we have performed western blot (WB) experiments to measure the abundance of CRBN under various conditions: 1 h, 3 h, and 24 h at both 1 μ M and 10 μ M concentrations (**Supplementary Fig. 9**). Our finding confirmed that MRT-31015 treatment results in reduced CRBN degradation compared to MRT-31619 while MRT-30568 completely loses degradation activity. Furthermore, no hook effect or CRBN recovery at high concentrations was observed for MRT-31015 and MRT-30568. These data are consistent with and further supported by CRBN-HiBiT experiments at multiple concentrations for all four compounds (**Fig. 1d-e, and Fig. 3g**). The main text has been revised to incorporate these additional experiments in the manuscript (line 184-187).

2) The authors should describe how they produced the sample for cryo-EM grid application. If the protein sample was taken directly from the SEC-MALS experiment described in the Methods, this should be specified.

Response: Great catch. We have updated the Method section “**Cryo-EM sample preparation and data acquisition**” with additional details on our cryo-EM grid sample preparation (line 506-510).

The cryoEM protein sample differed from the SEC-MALS sample. For cryoEM, we used Strep-TEV tagged CRBN⁴¹⁻⁴⁴² and untagged DDB1^{ΔBPB} (amino acids 1-395-GNGNSG-706-1140). For SEC-MALS, we used Strep-TEV tagged CRBN⁴¹⁻⁴⁴² and untagged full-length DDB1. The Methods section “**Recombinant protein expression and purification**” has also been updated accordingly (line 484-486).

3) Could the authors comment on any interactions between the two molecules of MRT-31619 bound in the complex??

Response: Excellent point. We have created three additional visualization figures in the main **Figure 2**. **Fig.2b** zoom-in at the tri-tryptophan CRBN pockets, **Fig. 2c** and **2g** are illustrating the interactions involving MRT-31619-MRT-31619 and CRBN-MRT-31619.

The main text has been revised to incorporate these additional Figures in the manuscript, and we have provided comments on the interactions between the two molecules of MRT-31619 (Line 134-146).

4) In line 82, the authors describe the in vitro complex between MRT-31619 and CRBN-DDB1 as stable, however the stable character of this complex is not fully described by this single measurement at a single concentration. At 9 μM CRBN-DDB1 and 10 μM MRT-31619, only partial dimerisation of CRBN-DDB1 is observed. The authors could consider refraining from making claims on stability in this case.

Response: Great point. We have removed “stable” in the corresponding sentence, in line 103-104.

5) To further characterise the ability of MRT-31619 to mediate dimerisation of CRBN, compared to the CRBN homoPROTAC 15a previously published by Steinebach et al. in 2018, and compared to the chemical analogues MRT-31015 and MRT-30568 published in this study, the authors could consider performing further biochemical

or biophysical characterisation. Several examples of assays are described in Maniaci et al. Nat. Commun. 2017 (DOI: 10.1038/s41467-017-00954-1), Diehl et al. Angew. Chem., 2024 (DOI: 10.1002/anie.202319456) which should be cited. Given that more stable ternary complexes with degrader molecules have been shown to drive degradation efficiency and potency, it would be interesting to understand here if CRBN dimerisation correlates with degradation efficacy.

Response: We appreciate the reviewer's recommendations regarding the further characterization of CRBN homodimerization through biochemical or biophysical methods. Indeed, we have conducted additional biophysical and biochemical characterization of MRT-31619 and the analogues, MRT-31015 and MRT-30568.

To evaluate the formation of the CRBN:MGD:CRBN ternary complex in solution, we performed Flow-induced Dispersion Analysis (FIDA), as shown in **Fig. 3i** and confirm that the extent of biophysical ternary complex formation correlates with the cellular potency of degradation. Furthermore, we utilized a biochemical thalidomide CRBN displacement assay to measure the binary affinity to CRBN for MRT-31619, MRT-31015, MRT-30568, and CRBN homo-PROTAC. The IC_{50}/K_i values are now reported in **Supplementary Fig. 2**. We have revised the manuscript to incorporate these additional experiments (line 180-183 and 189-199) and included the recommended references (line 73).

While we also performed Surface Plasmon Resonance (SPR) experiments, the likely dimerization of CRBN in solution in the presence of the glue in addition to the potential dimerization of two immobilized CRBN on the surface complicated the observed equilibria beyond a glue-mediated dimer formed by CRBN in solution with immobilized CRBN. Fitting such data for potentially multiple different binding scenarios would be ambiguous.

6) It is unclear whether the maps used for validation are deepEMhancer or RELION post-processed maps. It is a widely accepted opinion that the modified densities from density-modifying AI model programs should be primarily treated as modelling aids. The reviewers encourage the authors to perform any final atomic refinements and validation processes using volumes which have not been modified with AI-based procedures such as deepEMhancer.

Response: Details of model refinement are given in the **Methods**. The model was refined (and thus validated as implemented in phenix.real_space_refine) exactly as the reviewers suggest. DeepEMhancer maps were used as modeling aids and for illustration, shown

together with Relion Post-processed maps in **Supplementary Fig.4**. All maps were deposited (deepEMhancer maps as ‘additional maps’) in the EMDB.

We added the underlined sentence to the **Methods** for additional clarification (line 552-557):

(...) the model was prepared for refinement with phenix.ready_set and then sent through phenix.real_space_refine, first against the unsharpened map (refining ADPs) and then against the sharpened map from Relion (without ADP refinement). Model validation was performed against the sharpened map. All maps and models were deposited in the EMDB and PDB under accession codes EMDB-52330/PDB-9HPI (conformation 1) and EMDB-52331/PDB-9HPJ (conformation 2).

7) Could the authors specify which atomic models were used to generate the alignment for ubiquitination modelling in Extended Data Fig 10? From our own modelling using cryo-EM structures of Cul4A-DDB1 in the presence/absence of an E2 (PDB codes: 8B3G and 7OKQ, aligning DDB1 with the cryo-EM structures in this study), several lysine residues from the ubiquitinomics results shown in Fig. 1(b) on the trans CRBN protomer appear to be well-poised for ubiquitination, even in the presence of another molecule of DDB1. No lysines on DDB1 were shown to be ubiquitinated in this system, this is consistent with our modelling which shows that DDB1 is outside of the reach of E2-Ub. Therefore, we disagree with the statement in line 129 that the second CRBN protomer is devoid of DDB1. We would also like to highlight that in Extended Data Fig 10, the binding site denoted for the E2 is likely to be incorrect, as shown by several published X-ray crystallography and cryo-EM structures of cullin RING and other RING E3 ligases engaging ubiquitin-loaded E2s.

Response: Using the proposed E3/E2/CUL4/DDB1 model (PDB: 8B3G) it does not seem that the lysines of DDB1 are less accessible to the E2 than the most ubiquitinated lysines in CRBN (K300, K211, K226, K222). Indeed, we find that access to the CRBN lysines is partially occluded by DDB1 (see new **Supplementary Fig. 11**). Multiple lysines in DDB1 seem more accessible to the E3/E2 machinery than the most ubiquitinated lysines in CRBN, which as seen in new Supplementary Fig. 11 are partially occluded.

However, we agree with the reviewer that, based on the current data, alternate interpretations remain possible. In our revised manuscript, we present the ubiquitination data and the location of the lysines in new **Fig. 4**. In the main text, we propose possible explanations, both the possibility that the E2 cannot reach DDB1 as suggested by the reviewer, as well as the possibility that CRBN is present in the unbound apo state. New **Fig. 4** now presents the ubiquitination data (previously Fig. 1b) and a model of the location of

these lysines, along with a modeled DDB1 on the neosubstrate. In new **Supplementary Fig. 10** we present the DDB1 ubiquitinomics data, and in new **Supplementary Fig. 11** we present a model using one of the E3/E2 complexes suggested by the reviewer (PDB: 8b3g).

8) This seems a very short article, with unusually only two figures. It feels a little sparse and the narrative could expand. Also many panels from the SI could be moved to main text. the paper seems to have been written in a rush, with narrative jumping up and down, e.g. ubiquitinomics data presented early, then referred to it later on when introducing the model? It is suggested that many figure panels and data to be moved to the main text, for example: Compounds chemical structure of key compound(s) should be included in main text figure Comparison with CRBN Homo-PROTAC degrader 1 (also called compound 15a in the original publication) should feature more prominently upfront in the main text.

Response: Thank you for the suggestions. We have revised the manuscript and expanded the narrative (from 1227 words to 2744 words). Data from the Supplementary figures (Compound structure, Ubiquitinomics, Proteomics, HiBiT-CRBN, Nanobret and model) have been moved to the main figures and additional figures from new experiments were added. Ubiquitinomics data is now in **Fig.4** with the model. Compound chemicals are included in **Fig.1** and **Fig.3**. We have also expanded the abstract, results, and discussion sections.

9) use of scholar language is encouraged throughout - e.g. use of word "catapulted" seems excessive.

Response: Great point. We have replaced "catapulted" with "advancement", in line 52 and language has been revised throughout the document.

10) "...there are minimal protein-protein interactions between the two CRBN protomers" - could quantify this? for example by calculating buried SASA within the ternary complex (see Cowan & Ciulli Ann. Rev. Biochem 2021 (PMID: 35320687, currently cited as ref. 3); and PMID: 33930325). on this point - the NanoBRET data with PPI mutations is important and should be quantified.

Response: Thank you for the suggestions. We have created an additional visualization figure in the main **Fig. 2d** to show the footprints (i.e., the regions of the CRBN neosurface buried upon complex formation) for CRBN-MRT-31619: protomer 1 and protomer 2. We calculated both the solvent excluded surface (SES, also known as molecular surface) for MRT-31619

and CRBN and the solvent accessible area. The calculation of the solvent excluded surface allows us to visualize the buried area upon complex formation and attribute this area more easily to compound or PPI. The formation of the complex results in the burial of 1528 Å² of the solvent accessible surface area of both protomers (computed using freesasa). We have revised the manuscript to incorporate this additional figure (line 146-153)

For the NanoBret data with PPI mutants, we have calculated their corresponding EC₅₀, as shown in **Supplemental Fig. 8**. Despite the different mutation constructs, dose-response recruitment curves are still observed, and the EC₅₀ values do not clearly indicate a preferential PPI mutant. In order to further explore residues that may be involved in the interaction, based on the footprints, we further explored the role of V388, E377, and F102, residues that in mice CRBN (mCRBN) have a distinct role from human CRBN. We find that mouse CRBN does not homodimerize nor lead to degradation with MRT-31619. However, and surprisingly, mutating either S105F (F102 in hCRBN) or V388I/E377K restores binding partially or fully.

11) "...our data suggest a model where one CRBN protomer is part of a fully assembled CRL4 complex whereas the second protomer (the neosubstrate) is devoid of DDB1." this seems inconsistent with the SEC-MALS data which is consistent with a homo-dimeric complex of 2x CRBN-DDB1. It seems hard to imagine CRBN protein being stable alone and without being bound to DDB1.. what does a model of 2x full-length CRL4-CRBN look like?

Response: Thanks for your comments. Indeed, in our biochemical and biophysical experiments with recombinant proteins (SEC-MALS and cryo-EM), we do not observe DDB1-free CRBN. However, CRBN can exist in its unbound state, even if it is rapidly degraded by the cellular machinery¹, and our own experimentation shows that a CRBN incapable of engaging DDB1 in cells (hCRBN Trunc in **Fig. 2e**, **Supplementary Fig. 8**) retains its capacity to engage the either wildtype or truncated CRBN. To address this comment and comment #7, in the revised version of the manuscript, we mention that DDB1-free CRBN is only one of several possibilities for little DDB1 ubiquitination and lack of degradation (line 214-222). In revised **Fig.4**, we show the model of 2 full-length CRBN-DDB1 where we have mapped the ubiquitin lysine sites identified by ubiquitinomics, while in **Supplementary Fig. 10** we show the ubiquitinomics data for DDB1, and in **Supplementary Fig. 11** we show a model of the E3/E2 machinery with the DDB1/CRBN homodimer.

12) reference list if OK, though could be expanded. For example, the authors might

want to acknowledge and cite the original first homoPROTAC paper which was on VHL homo-dimerizers (PMID: 29018234)

Response: Thanks for your recommendations. We have included the VHL homoPROTAC papers (Maniaci et al. Nat. Commun. 2017 and Diehl et al. Angew. Chem., 2024, in line 73) and extended the list of references from 39 to 60.

13) EDF10: why not mapping all the K site found ubiquitinated in the MS spec ubiquitinomics experiment?

Response: We have mapped all ubiquitin lysine sites identified in the ubiquitinomics experiment onto the model shown in **Fig. 4**.

Minor points:

8) Page 1 line 49. We suggest re-wording "...these compounds bind to the CRBN CULT domain", as this makes it seem as though all MGDs and PROTACs recruit and bind the CRBN CULT domain.

Response: We have re-phrased it as ... many such compounds bind to the CRL4-CRBN ubiquitin ligase and promote recruitment of neosubstrates for ubiquitination and degradation., in line 54-55.

9) Differences in nomenclature 'HiBit-CRBN' and 'CRBN-HiBit' make it unclear whether the HiBit is on the N- or C-terminus of CRBN.

Response: Great catch. We used HEK293T CRBN KO + PGK-HiBit-CRBN cell line and therefore have HiBit on the N-terminus of CRBN. We have modified the text, methods and legend accordingly.

10) In Extended Data Fig. 4, the identity of 'Cpd-1' in the figure legend should be indicated as MRT-31619. This should also be reflected in the Methods (line 429).

Response: We have changed "Cpd1" to "MRT-31619" in **Supplementary Fig.3** and in the Methods, line 457.

11) Page 2 line 82. "Dimeric CRBN-DDB1 dimer" should be rephrased.

Response: We have removed "dimeric" in the sentence (line 103).

12) In Extended Data Fig 5, could the authors briefly comment on what the composition of the grey-coloured volume was which constituted 25% of the 3D classification results?

Response: Good spotting by the reviewers. You can find this figure in **Supplementary Fig.4**. These particles refined to a reconstruction of worse resolution (3.3 Å) roughly similar to conformation 1 (described in this study). As no new insights are gained from this reconstruction, a detailed description was omitted from this manuscript to increase readability. A sentence was added to the methods (line 543-545):

“A third class (resembling conformation 1) was identified at this step. Particles contained in this class led to a worse overall map with no additional insights, and it was thus not followed up in detail.”

13) The FSC curved in Extended Data Fig 5 (b) and (c) are difficult to read.

Response: Readability was increased, and plots were moved to **Supplementary Fig.5**.

14) From the viewing distribution, it appears that the two structures solved have some preferential orientation, and thus the global and local resolutions by FSC could be inflated. The directional FSCs, as well as the sphericity value, should be included for both structures and the authors should consider using the 3D FSC server for FSC plots: <https://3dfsc.salk.edu>

Response: 3DFSC plots and sphericity values are now included in **Supplementary Fig.5** and in **Supplementary Table 1**, respectively.

15) For cryo-EM data validation, the authors should consider including all defocus values used, adding the 3DFSC sphericity value and EMRinger score.

Response: The defocus is given as a range (-0.8 – -2.0 μm) in Methods and in the **Supplementary Table 1**, as is usual in the cryo-EM field. The discrete values used during data collection (in EPU/SerialEM) are normally not given, since they will not be precisely recapitulated in the acquired movies.

3DFSC sphericity and EMRinger score were added to the **Supplementary Table 1**.

16) Could the authors highlight the presence or absence of any symmetry in the MRT-31619 dimer assembly?

Response: No symmetry was imposed during image processing. In the **supplementary data, movie 1** shows pronounced flexibility of the complex, negating all potential overall C2 symmetry. The two moieties MRT-31619 inside the dimer assembly overlay almost perfectly with each other. There is no considerable asymmetry between the two compounds, but also no perfect symmetry.

We added the following sentences to the manuscript:

No symmetry was imposed during image processing, as substantial conformational flexibility was identified in the complex (Supplementary movie 1), line 118-120.

And:

Surprisingly, the two copies of MRT-31619 adopt a complementary helix-like conformation that bridge the two CRBN tri-Trp pockets (Fig. 2b). The conformation of the two compound copies is virtually identical within an all-atom RMSD of 0.064 Å, line 130-133.

We agree to waive our anonymity in a spirit to enhance the transparency of the peer review process, Alessio Ciulli & Charlotte Crowe.

Response: Thanks very much for the transparency, we really appreciate it.

Reviewer #2 (Remarks to the Author):

Reviewer #3 (Remarks to the Author):

This manuscript reports the discovery of a small molecule that induces dimerization and in turn degradation of cereblon by acting as a homodimer-inducing molecular glue. The authors discover the compound as selectively targeting cereblon by using proteomic profiling and present data in Jurkat cells showing high selectivity. The mechanism of interaction is resolved by using cryoelectron microscopy to solve the structure of cereblon with the small molecule (MRT-31619) and DDB1. The

conclusions are supported by the data and manuscript well-written. Only two small suggestions are made below.

Response: We thank the reviewer for the positive comments and appreciate the feedback.

Q1: A figure should be included that shows the interactions between MRT-31619 and the tri-Trp pocket, as described at line 102-104.

Response: Great point. We have created an additional visualization figure in the main **Figure 2. Fig. 2b** is showing a zoom-in view of the tri-Trp pocket, highlighting the interactions between MRT-31619 and the tri-Trp pocket.

Q2: It's not clear what the authors mean at line 135; are they proposing 'analogous pathway' to MRT-31619 induced dimerization and if so, how would that be induced endogenously?

Response: We appreciate the reviewer's attention to the sentence in line 135. For clarity, we have decided to remove it. Instead, we proposed discussing the opportunity to use MRT-31619 as chemical knockout tool molecule to investigate CRBN's endogenous substrates and physiological roles and elucidating the molecular mechanism of MGDs (line 238-253).

Reviewer #4 (Remarks to the Author):

In this manuscript Langousis and coworkers reported a molecular glue MRT-31619 that mediates the dimerization and degradation of CRBN. Intriguingly, two MGD molecules assemble into a helix-like structure and connect two copies of CRBN together. This is an interesting and unique finding in MGD design. This study included detailed biochemical, structural and cellular experiments to demonstrate the binding mode of MRT-31619 to CRBN. The results and conclusion are convincing. Given the important role of CRBN in targeted protein degradation, the report of MRT-31619 as a new CRBN molecular glue is well justified. Overall, this manuscript is well-written and concise, and should be of interest to a broad range of readers of the journal.

Response: We thank the reviewer for the positive comments and appreciate the feedback.

Q1: What is the binding affinity of MRT-31619 to CRBN? This information may be obtained using SPR or BLI binding assays using the small molecule and recombinant proteins.

Response: We appreciate the reviewer's recommendations regarding the further characterization of CRBN homodimerization through biochemical or biophysical methods. Indeed, we have conducted additional biophysical and biochemical characterizations of MRT-31619 and the analogues, MRT-31015 and MRT-30568.

To evaluate the formation of the CRBN:MGD:CRBN ternary complex in solution, we performed Flow-induced Dispersion Analysis (FIDA), as shown in **Fig. 3i**. Furthermore, we utilized a biochemical thalidomide CRBN displacement assay to measure the binary affinity to CRBN for MRT-31619, MRT-31015, MRT-30568, and CRBN homo-PROTAC. The IC_{50}/K_i values are now reported in **Supplementary Fig. 2**. The CRBN IC_{50} of MRT-31619 is at 14 nM and K_i of MRT-31619 at 7 nM. We have revised the manuscript to incorporate these additional experiments (line 180-183 and 189-199).

While we also performed Surface Plasmon Resonance (SPR) experiments, the likely dimerization of CRBN in solution in the presence of the glue in addition to the potential dimerization of two immobilized CRBN on the surface complicated the observed equilibria beyond a glue-mediated dimer formed by CRBN in solution with immobilized CRBN. Fitting such data for potentially multiple different binding scenarios would be ambiguous.

Q2: Figure 2 should be improved to show more details of the MRT-31619 binding to CRBN by including more residues interacting with the small molecule.

Response: Excellent point. We have created two additional visualization figures in the main **Figure 2**. **Fig.2d** and **Fig.2g** are highlighting the MRT-31619 binding to CRBN by including the residues interacting with MRT-31619. The main text has been revised to incorporate these additional Figures in the manuscript (Line 154-157).

Q3: Given that the two CRBN CULT domains demonstrated minimum direct interactions when being brought together by two MRT-31619 molecules, how rigid is the ternary complex of two CULT domains linked by the small molecules? Do the cryoEM images show particles with different conformations?

Response: Supplementary movie 1 shows the pronounced conformational flexibility of the complex, and we now refer to it when describing the structure (line119-120):

No symmetry was imposed during image processing, as substantial conformational flexibility was identified in the complex (Supplementary movie 1).

Q4: In the Extended Data Figure 9b, treatment with MRT-31619 led to degradation of CRBN. However, MRT-30568 treatment led to increase in CRBN level. Why?

Response: We thank the reviewer for identifying these points in the previous Extended Data Figure 9b. In the revised **Fig. 3h**, we agree that the HiBiT-CRBN curve tends to increase at higher concentrations of MRT-30568 and have removed the fitted line. We believe this is due to the variability of the HiBiT assay, and that the data at 10 μ M is an outlier data point. Proteomics analysis confirmed that the abundance of CRBN did not change after 24 h of treatment with 10 μ M MRT-30568 (**Fig. 3f**). To further support the Proteomics data, we performed Western Blot experiments to quantify the abundance of CRBN for MRT-30568 (**Supplementary Fig. 9**) under the following conditions: 1 h, 3 h, and 24 h at both 1 μ M and 10 μ M concentrations. These experiments confirmed that MRT-30568 does not lead to an increase in CRBN levels at higher concentrations.

Q5: In the model of CRL4CRBN complex bound with MRT-31619 as shown in the Extended Data Figure 10, the ubiquitination sites on the neosubstrate CRBN2 are facing away from the E2 binding site on CRL4. How would the charged E2 transfer the Ub to the lysine sites on CRBN?

Response: This is a great point, indeed. The location of the most ubiquitinated lysines, nearly-occluded by DDB1, and in opposite faces made us speculate that CRBN is preferentially ubiquitinated in an apo-state, possibly in a highly dynamic or molten-globule state, enabling the ubiquitination of lysines in opposite faces. However, as stated in response 7) and 11) to reviewer #1, we believe that characterizing this would require further studies. For this reason, we now present the ubiquitination data, along with a model for the lysine locations in the complex in new **Fig. 4** and discuss two possible explanations for this behavior. We also present an updated model of the full E3/E2 complex in new **Supplementary Fig. 11**.

Q6: What is the application of the MRT-31619 in targeted protein degradation if it removes CRBN from the treated cells. This should be discussed in the manuscript.

Response: We have expanded the discussion in the manuscript (224-267 line) to include arguments on the potential applications of MRT-31619 in targeted protein degradation.

Reference:

1. Song T, *et al.* CRL4 antagonizes SCFFbxo7-mediated turnover of cereblon and BK channel to regulate learning and memory. *PLoS Genet* **14**, e1007165 (2018).

Response to reviewers for “A degron-mimicking molecular glue drives CRBN homo-dimerization and degradation” (ID#/NCOMMS-25-15248-T)

We're grateful for the positive feedback from all four reviewers and their recommendation to publish in *Nature Communications*.

Reviewer #1 (Remarks to the Author):

The authors have done an excellent good job at revising the manuscript. The degradation data has been strengthened by performing additional western blot experiments, and more robust comparison on degradation efficacy and mode of action in comparison to the previously published CRBN homoPROTAC has been brought up-front in the manuscript. The new compound has also been compared with the previously published CRBN homoPROTAC in terms of in-solution binding by TR-FRET, monitoring a fluorescently-labelled thalidomide probe displacement. We previously highlighted that information seemed unclear in terms of cryo-EM sample preparation, and the information has now been added to the manuscript. No changes have been made to the cryo-EM data, but the results have been further dissected by highlighting key interactions, interfaces and buried surfaces. Data on how the well-known human to mouse species residues variations impact on TC formation and compound-induced CRBN degradation has been added. Some further discussion has been added for the ubiquitinomics passage albeit the conclusions from this section remain somewhat limited. However we view this as acceptable to be included as is, with the due caveats and limitations.

Overall, the paper is significant and reports an exciting unexpected and relevant finding. We previously made several suggestions, and we are satisfied that these have been addressed. In the process of strengthening the manuscript, several sections have been altered, and additional experiments have been performed, meriting another complete read-through of the latest manuscript version. We believe this version is much improved and we recommend publication in the journal, subject to some additional and final revisions, as outlined below:

1) Line 75: and Supplementary Figure 2: for the ‘biochemical thalidomide displacement assay’ please specify in the main manuscript and in the figure legend that this is a TR-FRET assay between CRBN and thalidomide.

Response: Excellent point. We have revised the wording on line 76/77 and legend of Supplementary Figure 2.

2) Supplementary Figure 2: does Rh refer to hydrodynamic radius? If so, is this a usual readout for a TR-FRET assay?

Response: Great catch. We have updated the y axis label in Supplementary Figure 2 to “% Displacement”.

3) Line 108: based on the data shown in Fig. 1g, it does not appear that the CRBN-CRBN interaction is fully abrogated in the dual CRBN W386A condition (green curve). In contrast the HaloTag-CRBN(W386A) with CRBN(WT)-NanoLuc does not yield any increase in BRET ration upon compound treatment. Have these been mislabelled in the plot?

Response: Great catch. We have updated the legend of Figure 1g. We mislabeled “HaloTag-CRBN_WT + CRBN-Nluc_W386A” with “HaloTag-CRBN_W386A + CRBN-Nluc_W386A”.

4) The authors state in Supplementary figure 10 that DDB1 is not ubiquitinated in the presence of MRT-31619, however in the main manuscript they state that DDB1 K823 showed significant ubiquitination. Given the stability of the CRBN-DDB1 complex, it seems unlikely that CRBN would exist without DDB1. Several other factors could come into play, such as CRBN conformational change in the presence of MRT-31619 and another CRBN molecule versus its apo form in the presence of DMSO, and also flexibility of CUL4 in the presence of two units of CUL4A-E2-Ub which would create steric hindrance and thus adapt the ‘light face’ of accessible lysines to the E2 active site cysteine.

Response: The title of Supplementary Figure 10 has been revised to “Ubiquitinomics data labeling the DDB1 ubiquitinated peptides.” replacing the original “DDB1 is not ubiquitinated upon MRT-31619 treatment”. We agree with the reviewer that multiple possibilities could explain the lack of DDB1 degradation. We have added the following comment to the end of the ubiquitinomics section (line 221-223): “In addition to these two hypotheses, several

other factors, such as E3/E2 dynamics and protein stoichiometry, could help explain the lack of DDB1 degradation.”

5) In Supplementary figure 11. For clarity, the authors could consider colouring the lysine residues which showed evidence of ubiquitination in a different colour. Can any clusters of ubiquitinated lysines on CRBN be highlighted, as has been shown for example in Fig 2 (B) DOI: 10.1126/sciadv.ado6492?

Response: Thanks for the great suggestion. In the new Supplementary Figure 11, we have colored the lysines which are not ubiquitinated in pink and ubiquitinated lysines in yellow. We have also zoom-in the area with the most significant upregulated ubiquitinated lysines and show them in a different representation for clarity.

6) When discussing the qualification and potential utility of their compound as a CRBN degrader, the authors should duly acknowledge that MRT-31619 is not the first CRBN-degrader that has been developed in the field, and thus duly reference the previous studies, not only the CRBN homo-PROTAC study (that is already cited in the manuscript as reference 13), and the RSC Med Chem 2021 (also cited as ref. 14), but also the papers that reported the VHL-CRBN ligase vs ligase PROTACs, which were all found to induce degradation of CRBN and not VHL – specifically, Girardini et al. Bioorg Med Chem, 2019 (PMID: 30826187), and Steinebach et al. Chem Commun 2019 (PMID: 30672516).

Response: We have revised lines 72-74 and lines 244-246 and added two references related to the VHL-CRBN PROTAC.

Minor points:

7) Line 70: ‘During proteomic profiling of our compound CRBN library’ should be rephrased, for example ‘CRBN-focused compound library’ or ‘CRBN-recruiting compound library’

Response: We have revised the text line 70 to “our CRBN-focused compound library”.

8) In Figure 1, the authors adequately explain in the figure legend what each abbreviation means (e.g. btzb, len, MLN, HT, etc.). However, there seems to be ample

space to write out most of the full term below the figures themselves. This would be easier to understand, particularly for those outside of or new to the TPD field.

Response: We have updated the Figures and legends for Figure 1d, 1e, 1f and 1g to include the full names of Bortezomib, Lenalidomide, MLN4924 and HaloTag.

9) Some inconsistencies when referring to the CRBN homoPROTAC 15a: CRBN homo-PROTAC, CRBN-homoPROTAC, CRBN homoPROTAC and CRBN PROTAC are present in the main manuscript, figures and SI.

Response: We have revised the text to refer to “CRBN homo-PROTAC” in the Figure 1 and its legend, in Supplementary Figure 1 and its legend, Supplementary Figure 2 and in Supplementary Figure 7.

10) Figure 2e/f and Supplementary figure 8: the plots are difficult to read with all curves except the wild-type being in grey. Could the authors clarify in the figure legend the difference between hCRBN and mCRBN?

Response: We revised the color of the curves in Figures 2e and 2f, as well as in Supplementary Figures 8a–8c to improve visual clarity. Additionally, we have updated the legends for Figure 2 (line 832) and Supplementary Figure 8.

11) Supplementary figure 9 and lines 184-187: could the N = 2 replicate degradation western blots be quantified relative to GAPDH for clarity to the reader? For example, in the final lane (MRT-30568 10 uM 24 hours) the CRBN band appears to decrease in intensity, however GAPDH is also at lower intensity.

Response: We have included the quantification of CRBN normalized to GAPDH in Supplementary Figure 9. Below, we present data from the second biological Western blot replicate, which is not included in the manuscript.

12) Could the authors clarify or rephrase the following sentence, lines 210-213: Despite the proximity of most ubiquitinated lysine, only one DDB1 ubiquitinated lysine at K823 showed significant upregulation after MRT-31919 (Supplementary Fig.10), which was insufficient to drive DDB1 degradation, as shown by proteomics (Fig.1b)

Response: We have clarified this sentence as follows (line 212-215): “Although several lysine residues on DDB1 are in close proximity to CRBN, only K823 exhibited a significant

increase in ubiquitination following MRT-31619 treatment (Supplementary Fig. 10). However, this modification at K823 was insufficient to induce DDB1 degradation, as confirmed by global proteomics analysis (Fig. 1b).”

13) In the new Figure 4 (b), on the right-hand-side panel DDB1 has been written as “DDB”.

Response: We have updated Figure 4 (b).

We agree to waive our anonymity in a spirit to enhance the transparency of the peer review process, Alessio Ciulli & Charlotte Crowe.

Reviewer #2 (Remarks to the Author):

Reviewer #3 (Remarks to the Author):

This manuscript has been improved and my concerns have been addressed. One remaining concern is that there does not appear to be a PDB validation report for the deposited structures. It's possible it was present in the first submission and missing in the revision or that it's present and just eluding me.

Response: Thanks for your positive feedback. We have included the validation reports for the two deposited cryo-EM maps in the current submission (PDB 9HPI and PDB 9HPJ).

Reviewer #4 (Remarks to the Author):

The authors have addressed my points satisfactorily. The manuscript has been substantially improved and is now ready for publication.

Response: Thanks for your positive feedback